# Nucleus accumbens dopamine tracks aversive stimulus duration and prediction but not value or prediction error

Jessica N Goedhoop[1,2], Bastijn JG van den Boom[1,2†], Rhiannon Robke[1,2†], Felice Veen[1,2], Lizz Fellinger[1,2], Wouter van Elzelingen[1,2], Tara Arbab[1,2], Ingo Willuhn[1,2*]

[1]Netherlands Institute for Neuroscience, Royal Netherlands Academy of Arts and Sciences, Amsterdam, Netherlands; [2]Department of Psychiatry, Amsterdam UMC, University of Amsterdam, Amsterdam, Netherlands

**Abstract** There is active debate on the role of dopamine in processing aversive stimuli, where inferred roles range from no involvement at all, to signaling an aversive prediction error (APE). Here, we systematically investigate dopamine release in the nucleus accumbens core (NAC), which is closely linked to reward prediction errors, in rats exposed to white noise (WN, a versatile, underutilized, aversive stimulus) and its predictive cues. Both induced a negative dopamine ramp, followed by slow signal recovery upon stimulus cessation. In contrast to reward conditioning, this dopamine signal was unaffected by WN value, context valence, or probabilistic contingencies, and the WN dopamine response shifted only partially toward its predictive cue. However, unpredicted WN provoked slower post-stimulus signal recovery than predicted WN. Despite differing signal qualities, dopamine responses to simultaneous presentation of rewarding and aversive stimuli were additive. Together, our findings demonstrate that instead of an APE, NAC dopamine primarily tracks prediction and duration of aversive events.

**\*For correspondence:**
i.willuhn@nin.knaw.nl

†These authors contributed equally to this work

**Competing interest:** The authors declare that no competing interests exist.

## Editor's evaluation

The article by Goedhoop et al. provides an important analysis of the role of terminal dopamine release in the nucleus accumbens in processing aversive events that will be of value to researchers interested in the neural mechanisms of reinforcement learning and computational modeling of dopamine function. Using a variety of conditions, the authors provide convincing data in support of the role of accumbal dopamine release in processing aversive events that situate the current report among growing interest and mounting investigations into the role of dopamine in aversion.

## Introduction

The midbrain dopamine system plays critical roles in motivation, learning, and movement; specifically for learning about rewards and creating motivational states that promote reward-seeking (*Berridge and Robinson, 1998*; *Bromberg-Martin et al., 2010*; *Berke, 2018*; *Schultz, 2019*). One of the most prominent functions of dopamine is the encoding of a so-called reward prediction error (RPE) signal (*Schultz et al., 1997*): when a reward is fully predicted by a cue, the increase in dopamine cell firing and terminal release of dopamine shifts 'backward' in time from the moment of reward delivery, to that of cue presentation (*Schultz et al., 1997*; *Flagel et al., 2011*). Furthermore, dopamine neurons pause their firing when a predicted reward is omitted and increase their firing in response to the delivery of an unpredicted reward. Thus, dopamine neurons encode the difference between predicted

and obtained reward, which is corroborated by the fact that dopamine neuron activity scales with the relative value of reward and unexpected deviations from this value (***Bromberg-Martin and Hikosaka, 2009***).

Although the vast majority of studies focus on the relationship between dopamine and stimuli with a positive valence (rewards), the relevance of the dopamine system in processing stimuli with the opposite valence (aversive) has also generated great interest. In contrast to the primarily stimulatory response of rewards on dopamine activity, the reports on the effect of aversive events on the dopamine system are less consistent. For example, on the level of dopamine neuron cell bodies, aversive stimuli were demonstrated to result in inhibition of neuronal activity (***Ungless et al., 2004***; ***Mileykovskiy and Morales, 2011***), excitation thereof (***Anstrom et al., 2009***; ***Valenti et al., 2011***), or no effect at all (***Mirenowicz and Schultz, 1996***; ***Fiorillo, 2013***). The widely accepted explanation for these varying results is that subpopulations of dopamine neurons exhibit different response profiles to aversive stimuli (***Schultz and Romo, 1987***; ***Guarraci and Kapp, 1999***; ***Coizet et al., 2006***; ***Bromberg-Martin and Hikosaka, 2009***; ***Zweifel et al., 2011***; ***Cohen et al., 2012***; ***Lammel et al., 2011***), whereby variance is presumably introduced by different types of aversive stimuli, by the fact that some studies were performed in awake and others in anesthetized animals, and by the location and projection targets of the recorded dopamine neurons (***Brischoux et al., 2009***; ***Matsumoto and Hikosaka, 2009***; ***Lammel et al., 2011***). However, activity at the level of dopamine neuron cell bodies does not necessarily always translate to their projection targets (***Mohebi et al., 2019***) as axonal terminal release of dopamine may be capable of operating independently from cell body activity (***Threlfell et al., 2012***). Therefore, in interrogating the entire spectrum of functions of the dopamine system, it is imperative to include measurements of extracellular dopamine concentrations in the projection target.

Midbrain dopamine neurons modulate their targets via population signals: dopamine release from a large number of extrasynaptic terminals, combined, constitutes a diffusion-based signal that is perpetuated by volume transmission (***Rice and Cragg, 2008***). The vast majority of projections from dopaminergic neurons target the striatum and its subregions. Inconsistent with the classic hypothesis positing that the dopamine system broadcasts a uniform signal across the striatum, it has been reported multiple times in recent years that dopamine signals display regional heterogeneity (***Willuhn et al., 2012***; ***Willuhn et al., 2014a***, ***Lammel et al., 2011***; ***de Jong et al., 2019***; ***Menegas et al., 2017***; ***Klanker et al., 2017***; ***van Elzelingen et al., 2022a***; ***van Elzelingen et al., 2022b***). This heterogeneity is reflected in dopamine responses to aversive events throughout the striatum: whereas microdialysis studies report an increase in dopamine release in the nucleus accumbens in response to aversive events (***Young et al., 1993***; ***Young, 2004***; ***Wilkinson et al., 1998***; ***Bassareo et al., 2002***; ***Pascucci et al., 2007***; ***Ventura et al., 2007***; ***Martinez et al., 2008*** but see ***Mark et al., 1991***; ***Liu et al., 2008***), studies employing techniques with a higher, subsecond temporal resolution (e.g., fast-scan cyclic voltammetry [FSCV] or fluorescence fiber photometry) arrive at less consistent conclusions. For example, aversive stimuli produced an increase in dopaminergic activity in the nucleus accumbens shell (NAS) in some studies (***Badrinarayan et al., 2012***; ***de Jong et al., 2019***), but a decrease in others (***Roitman et al., 2008***; ***Wheeler et al., 2011***; ***McCutcheon et al., 2012***; ***Twining et al., 2015***). Similarly, contradictory findings are also reported in the neighboring nucleus accumbens core (NAC), where studies found both increased (***Budygin et al., 2012***; ***Mikhailova et al., 2019***; ***Kutlu et al., 2021***; ***Kutlu et al., 2022***) and decreased dopamine activity (***Badrinarayan et al., 2012***; ***Oleson et al., 2012***; ***de Jong et al., 2019***; ***Stelly et al., 2019***). In contrast, in the tail of the striatum, aversive events exclusively result in increased dopaminergic activity (***Menegas et al., 2017***; ***Menegas et al., 2018***). Overall, it can be concluded that most studies observe a change in dopaminergic activity in response to aversive stimuli, that there are substantial differences between striatal regions in this response, and that it remains unclear what determines whether aversive events provoke an increase or a decrease in dopaminergic activity within striatal regions.

Delineating the role of dopamine in processing aversive events crucially requires understanding what the above-described changes in dopamine signaling encode specifically; or, in other words, whether these changes reflect aversive prediction errors (APEs, in which the dopamine response would reflect the discrepancy between expected and received aversive events) or merely individual aspects of aversive conditioning (such as the presence of aversive stimuli, and/or their prediction). A thorough analysis by ***Fiorillo, 2013*** concluded that dopaminergic midbrain neurons do not encode aversive stimuli, but other studies have observed aspects of a dopamine APE, such as the predictive

cue adopting the dopamine response of an aversive stimulus (*Guarraci and Kapp, 1999*; *Oleson et al., 2012*; *Badrinarayan et al., 2012*), or an APE-like response when the aversive stimulus was unpredicted or omitted (*Matsumoto and Hikosaka, 2009*; *Matsumoto et al., 2016*; *Menegas et al., 2017*; *Salinas-Hernández et al., 2018*; *de Jong et al., 2019*). However, it should be kept in mind that in case of omission or early termination of an expected aversive event, rewarding aspects of a milder-than-expected aversive event (RPE) may be mixed with an APE (*Oleson et al., 2012*; *Salinas-Hernández et al., 2018*; *Stelly et al., 2019*).

To consolidate these contradictory findings, we systematically evaluated whether dopamine truly signals an APE through a series of behavioral experiments in rats, in which we varied the value of the aversive stimulus, context valence, and probabilistic contingencies, and compared aversive and appetitive conditioning. Using FSCV, we measured the NAC dopamine response to these conditions since the NAC is a hotspot for RPE-like signals (*Flagel et al., 2011*; *Papageorgiou et al., 2016*) and is also tightly linked to motivational processes related to aversion avoidance (*Badrinarayan et al., 2012*; *Oleson et al., 2012*; *Stelly et al., 2019*). We employed loud white noise (WN) as the aversive stimulus. WN possesses several merits as it is well-tolerated by rats and is not painful, precisely controllable (intensity and duration can be effortlessly titrated), aversive without inducing freezing (most pertinent to this study as this might interfere with the dopamine signal), does not jeopardize the recording equipment, does not introduce artifacts to the recordings, and can be administered reliably (see 'Discussion'). Based on the above-described findings, we hypothesized that NAC dopamine would exhibit an APE. However, we found that NAC dopamine concentration ramps down in response to both WN exposures and its predicting cue, ramps back upward upon stimulus cessation, that these ramps were qualitatively different from appetitive conditioning, and were inconsistent with a full APE signal.

## Results

### WN is aversive

We established the aversiveness of loud WN in a series of experiments. First, we used a real-time place aversion test in an open field, in which rats were exposed to 90 dB WN upon entry into one of the four quadrants of an open field (the WN quadrant was assigned randomly; *Figure 1A*). An example path traversed by a rat over the course of 30 min is depicted in *Figure 1A* (left panel), where the dark shaded area represents the quadrant paired with WN. On average, rats (n = 10) spent a significantly smaller percentage of time in the WN quadrant (9.71 ± 1.65) compared to chance level ($t(9) = 9.585$, $p<0.0001$; *Figure 1A*, right panel). Second, we exposed a new cohort of rats (n = 15) to the same open field without WN exposure to assess their preferred quadrant (*Figure 1B*, left panel). After this session, rats were placed in the open field in a second session, where now the entry into their preferred quadrant led to 90 dB WN exposure (*Figure 1B*, middle panel), which caused the rats to spend significantly less time in it (no WN: 48.5 ± 0.2474, 90 dB: 7.983 ± 0.3875, $t(14) = 13.43$, $p<0.0001$; *Figure 1B*, right panel). Next, a third cohort of rats (n = 12) was exposed to an open-field foraging task in which food pellets were placed between grid-floor bars in one of the quadrants. Entry into this grid-floor quadrant led to exposure of 0, 70, 80, 90, or 96 dB of WN, or a tone in separate sessions (*Figure 1C*, left panel). Without WN, rats spent 65.2% of the time in the maze foraging in the grid-floor quadrant. The greater the WN intensity, the shorter the amount of time rats spent foraging for food pellets ($\chi^2(5) = 27.6$, $p<0.0001$), whereas the tone did not reduce foraging (*Figure 1C*, middle panel), indicating that WN is more aversive than a pure tone ($t(10) = 2.389$, $p=0.0381$). The average number of entries into the grid-floor quadrant per session was 25.9 ± 2.5. The number of entries did not differ significantly across auditory stimuli. Rats were exposed to each auditory stimulus during foraging in two rounds of sessions, where they underwent exposure to all WN intensities and the tone in a first round of sessions, before exposure to each in a second round of sessions. The rats' response to the different WN intensities and the tone was reliable across first and second sessions (*Figure 1C*, right panel), demonstrating the absence of sensitization or habituation of the behavioral response to WN across sessions and days. Interestingly, exposure to WN stimulated locomotor activity: rats in an operant box significantly increased locomotion speed in response to semi-random presentation of 6 s WN bouts ($t(13) = 7.059$, $p<0.0001$; *Figure 1D*, middle panel). This locomotor response was WN-intensity dependent as we found a main effect of intensity

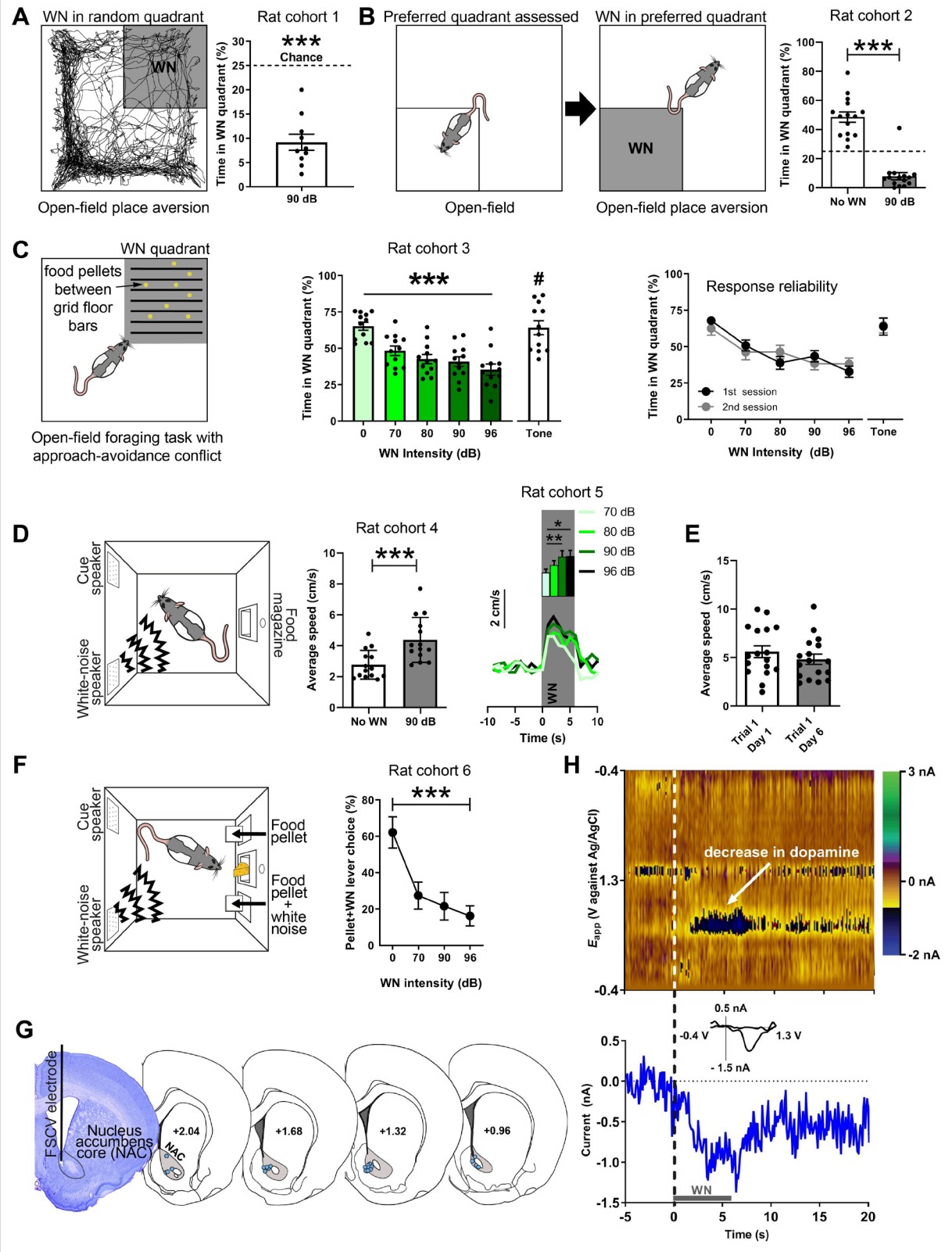

**Figure 1.** White noise (WN) is an aversive stimulus that lowers dopamine concentration in the nucleus accumbens core (NAC). (**A**) Left: example trajectory of a rat in the real-time place aversion test (30 min), in which entry into one quadrant (shaded), which was randomly determined prior to the session, led to 90 dB WN exposure. Right: aversiveness of WN in the place aversion test quantified as significantly decreased time spent in the WN quadrant (n = 10, 9.71 ± 1.65, t(9) = 9.585, p<0.0001). (**B**) Left: a second cohort of rats was placed in the open field to assess their preferred quadrant in

Figure 1 continued

a first session. In a second session, the entry into the preferred quadrant led to 90 dB WN exposure. Right: when WN was administered in this quadrant, rats spent significantly less time in it (n = 15, No WN: 48.5 ± 0.2474, 90 dB: 7.983 ± 0.3875, t(14) = 13.43, p<0.0001). (**C**) Left: a third cohort of rats (n = 12) was exposed to an open-field foraging task in which food pellets were placed between grid-floor bars in one of the quadrants. Entry into the grid-floor quadrant (shaded) led to exposure of 0, 70, 80, 90, or 96 dB of WN, or a 3 kHz tone in separate sessions. Middle: increasing WN intensity dose-dependently decreased the amount of time rats spent foraging for food pellets in the grid-floor quadrant ($\chi^2$(5) = 27.6, p<0.0001), whereas the tone did not reduce foraging (70 dB WN vs. 70 dB tone (#); t(10) = 2.389, p=0.0381). Right: the rats' response to the different WN intensities and the tone was reliable across first and second sessions of exposure, where rats underwent exposure to all WN intensities and the tone in a first session, before exposure to each in a second session. (**D**) Left: all other experiments took place in operant boxes equipped with a food magazine, a multiple-tone generator (cue speaker), and a WN generator (WN speaker). Middle: semi-random presentations of 6 s WN bouts increased the locomotion speed of rats (cohort 4; n = 14) in an operant box during the WN epoch compared to pre-WN baseline (post-hoc Dunn's test, t(13) = 7.059, p<0.0001). Right: in another cohort of rats (cohort 5; n = 13), we tested different WN intensities and found a main effect of intensity on locomotion speed ($\chi^2$(3) = 13.80, p=0.0032) and significant differences between 70 and 90 dB (p=0.005) and 70 and 96 dB (p=0.0143) (n=13). (**E**) Rats (n = 17) responded reliably with increased locomotion speed to WN across days. (**F**) Rats (cohort 6; n = 9) discern between different WN magnitudes in an operant choice task, where they had to choose between pressing a lever that resulted in a food-pellet delivery and a lever that resulted in a food-pellet delivery plus simultaneous 5 s of 0, 70, 90, or 96 dB of WN (pellet + WN; Friedman test, $\chi^2$(3) = 11.57, p=0.0003). (**G**) Left: example cresyl violet-stained brain slice depicting an electrolytic lesion in the NAC (outlined) at the tip of the fast-scan cyclic voltammetry (FSCV) electrode (vertical black line). Right: schematic overview of FSCV recording locations (blue dots) in the NAC (gray) of all animals. (**H**) Single-trial pseudocolor plot (top panel), dopamine trace (bottom panel), and cyclic voltammograms (inset in bottom panel) for representative, dopamine-specific current fluctuations recorded in NAC, 5 s before WN (dashed line), during 6 s of WN (gray bar), and 14 s after WN. Except for panel (**H**), data are mean ± SEM. *p<0.05, **p<0.01, ***p<0.001.

on baseline-subtracted locomotion speed ($\chi^2$(3) = 13.80, p=0.0032) in another cohort of rats (n = 13), and significant differences between 70 and 90 dB (p=0.005) and 70 and 96 dB (p=0.0143) (*Figure 1D*, right panel), which demonstrates that rats were able to discriminate between the different WN intensities. Rats responded with a comparable increase in locomotion speed to WN on days 1 and 6 of behavioral training, underlining the stability of the WN effect across sessions and days (t(16) = 0.9111, p=0.3757; *Figure 1E*). Next, we validated that rats can discern between different magnitudes of WN in an operant choice task. Here, rats could choose between pressing one of two extended levers, both of which prompted immediate delivery of a food pellet, but one of which additionally presented 5 s of 0, 70, 90, or 96 dB WN. Unsurprisingly, rats (n = 6) preferred the non-WN lever as indicated by a significant main effect of WN intensity using a Friedman test ($\chi^2$(3) = 11.57, p=0.0003). Importantly, when comparing the WN-paired lever presses, rats significantly preferred the 70 dB WN to the 96 dB WN (post-hoc Dunn's tests, p=0.0027; *Figure 1F*).

## WN suppresses dopamine release in the NAC

All FSCV recordings were conducted in operant boxes equipped with a food magazine, a multiple-tone generator (cue speaker), and WN generators (WN speaker) (Med-Associates; *Figure 1D*, left panel). Electrolytic lesions at the FSCV electrode tip were used to histologically verify electrode placement (*Figure 1G*, left panel). Histological analysis confirmed for all animals included in this study that the sensing end of their FSCV electrodes was consistently placed in the NAC (*Figure 1G*, right panel). Unexpected exposure to 6 s of 90 dB WN strongly and reliably decreased extracellular concentrations of dopamine in the NAC (*Figure 1H*).

## Different temporal NAC dopamine dynamics during aversive and appetitive Pavlovian conditioning

Dopamine release in the NAC is often consistent with a temporal difference RPE, where an increase in dopamine activity, initially time-locked to the delivery of a reward, shifts backward in time to its predicting cue. It is assumed that this phenomenon reflects the learned association between predictive cue and reward, where the reward becomes fully predicted by the cue and, therefore, no prediction error occurs at the time of reward delivery after sufficiently repeated cue-reward pairings (e.g., *Schultz et al., 1997*; *Flagel et al., 2011*). Our first experiment investigated whether a similar phenomenon also applies to aversive stimuli and their predictors, and in what time frame such a shift may occur. Rats (n = 16) were exposed to 30 pairings of cue (5 s) and 90 dB WN (6 s) that were separated by a variable intertrial interval (*Figure 2A*). In order to visualize the rapid changes in dopamine response presumably reflecting learning, the first five trials are depicted individually, and based on their stable visual appearance, trials 6–10 and 11–30 were binned together. Using one-way repeated-measures

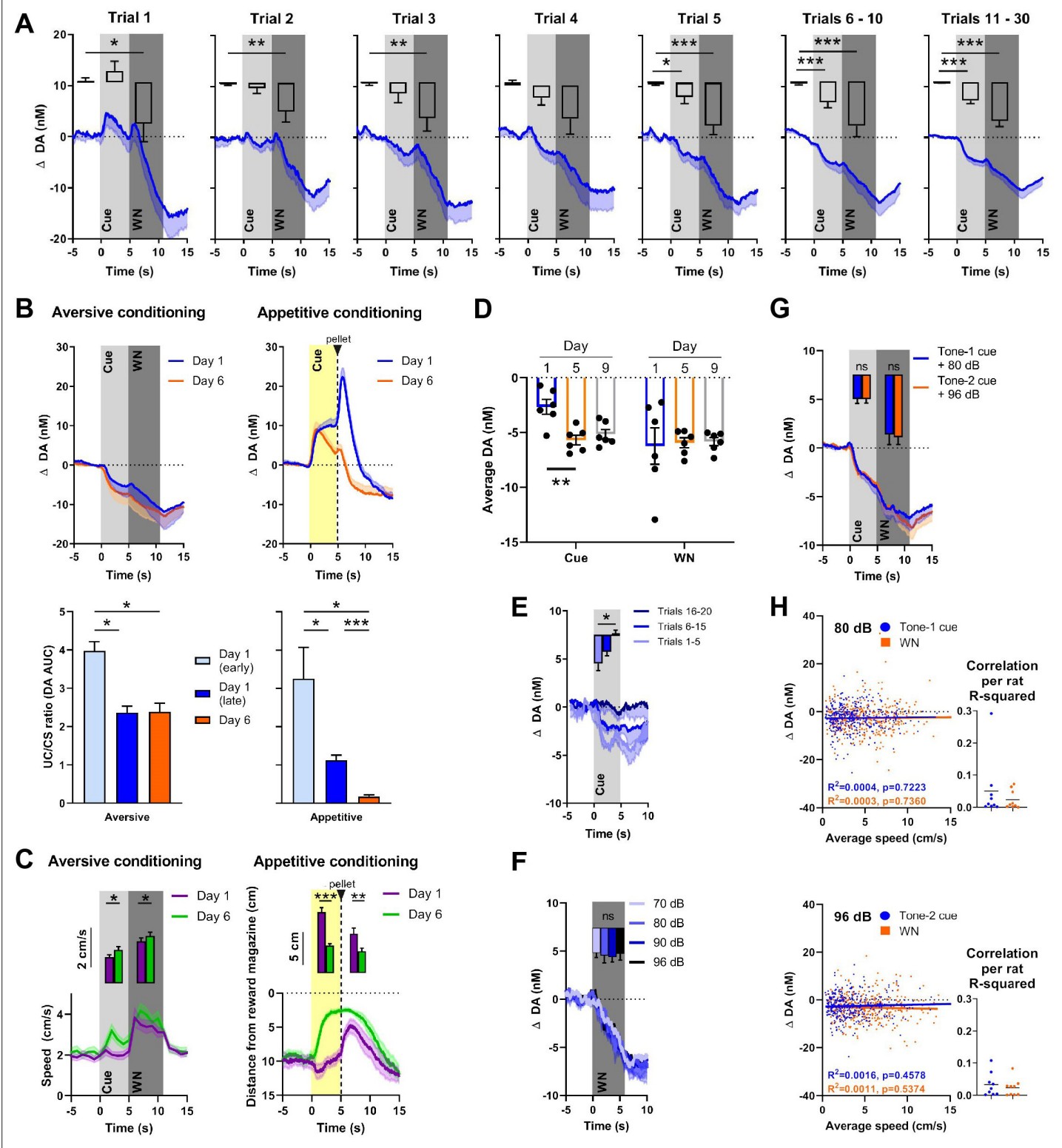

**Figure 2.** Nucleus accumbens core (NAC) dopamine signaling and rat behavior during Pavlovian white noise (WN)-cue conditioning and varying WN intensities. (**A**) Average extracellular concentrations of dopamine (DA; in nM) in the NAC (dark-blue line; SEM is shaded light blue) during the first 30 pairings of cue (5 s tone) and WN (6 s, 90 dB) (16 rats). To illustrate the immediate, unconditioned effects of WN, the first five trials are displayed individually. The bar graph insets depict dopamine release averaged for baseline, cue, and WN epochs. WN decreased dopamine significantly in all trials, except trial 4. The WN-paired cue began to decrease dopamine significantly starting at trial 5. (**B**) Comparison of dopamine release during

*Figure 2 continued on next page*

*Figure 2 continued*

aversive (left, n = 4) and appetitive (right, n = 10) Pavlovian conditioning. Top: subsecond changes in dopamine concentration (nM) on day 1 (blue) and day 6 (orange). Bottom: ratio of areas under the curve (AUC) between the CS and US (US/CS) on day 1 (early and late trials) and day 6 during aversive (left) and appetitive (right) conditioning. For aversive conditioning, dopamine differed between day 1 (early) and day 1 (late) (p=0.0138), and between day 1 (early) and day 6 (p=0.0318). For appetitive conditioning, a significant difference was found between early and late conditioning on day 1 (p=0.0441) and dopamine differed between day 1 (early) (p=0.0102) and day 1 (late) (p<0.0001) compared to day 6. (**C**) Conditioned behavioral response corresponding to (**B**). Left: during aversive conditioning, locomotion speed during cue presentation increased from day 1 to day 6 (Z = – 2.485, p=0.013), and also during WN (Z = –2.343, p=0.019) (n = 17). Right: during appetitive conditioning, time spent in proximity of the reward magazine increased between days 1 and 6 both during cue presentation (t(9) = 6.962, p<0.0001) and after pellet delivery (t(9) = 2.572, p=0.0301), (n = 10). (**D**) WN and its paired cue decreased dopamine concentration reliably across days. Cue-induced decrease was stable between days 5 and 9, and WN-induced decrease was stable between days 1, 5, and 9. (**E**) In an extinction session, for 20 consecutive trials, WN was withheld after cue presentation (n = 6), and dopamine differed significantly between trials 1–5 and 16–20 (p=0.0133). (**F**) In contrast, we detected no differences in dopamine release between exposure to varying WN intensities (70, 80, 90, or 96 dB; F(2.380, 11.90) = 0.1655, p=0.8813, n = 6). (**G**) We observed no significant differences in dopamine during cue (Z = –0.059, p=0.953) or WN (Z = –0.178, p=0.859) when two separate tones were used as predictors for 80 dB (blue) or 96 dB (orange) WN (n = 9). (**H**) Trial-by-trial correlation between locomotion speed and dopamine concentration during cue (blue) and WN (orange) for either 80 dB (top) or 96 dB WN (bottom) were not significant. Each dot represents one trial. Trials from all animals (n = 9) were pooled. Top left: no correlation during 80 dB WN ($R^2$ = 0.0016, p=0.457) or its cue ($R^2$ = 0.0004, p=0.7223). Bottom left: no correlation during 96 dB WN ($R^2$ = 0.0011, p=5374) or its cue ($R^2$ = 0.0003, p=0.7360). Right: $R^2$ values calculated separately for each individual rat confirm that there is no significant correlation between locomotion speed and dopamine. Data are mean ± SEM, + SEM, or - SEM. *p<0.05, **p<0.01, ***p<0.001.

ANOVAs, in which we compared the average dopamine concentration during baseline, cue, and WN epochs, we found significant main effects in all trials (trial 1: F(1.403, 19.65) = 6.853, p=0.0102; trial 2: F(1.324, 19.86) = 8.205, p=0.0059; trial 3: F(1.265, 18.98) = 6.737, p=0.013; trial 4: F(1.100, 16.50) = 5.016, p=0.0363; trial 5: F(1.548, 23.23) = 21.90, p<0.0001; trials 6–10: F(1.025, 15.38) = 15.94, p=0.0011; trials 11–30: F(1.093, 16.39) = 38.42, p<0.0001). Post-hoc analyses using Wilcoxon signed-rank tests with a Holm–Bonferroni multiple-comparison correction revealed significantly lower dopamine concentrations during the WN epoch compared to pre-cue baseline during trial 1 (Z = –1.988, p=0.0235), trial 2 (Z = –2.430, p=0.0075), trial 3 (Z = –2.327, p=0.010), trial 5 (Z = –3.464, p<0.0005), trials 6–10 (Z = 3.103, p=0.001), and trials 11–30 (Z = –3.516, p<0.0001), indicating that the decrease in dopamine during WN is an unconditioned response, since it is observed already during the first trial. In contrast, we only observed significantly lower dopamine in the cue epoch compared to pre-cue baseline during trial 5 (Z = –1.965, p=0.0245), trials 6–10 (Z = –2.999, p=0.0015), and trials 11–30 (Z = –3.516, p<0.0001), indicating that this decrease develops over time and, therefore, is a conditioned response. In these first 30 trials, the decrease in extracellular dopamine during the WN epoch did not disappear or decrease. Thus, WN does not provoke a substantial temporal shift of NAC dopamine signaling from unconditioned to conditioned (predictive) cue within the first session of training.

One possible explanation for an incomplete shift of dopamine signaling from WN to cue is that 30 pairings are insufficient to fully acquire the association. Therefore, we conditioned a subset of rats (n = 4) for five additional days. Another group of rats (n = 10) received food-pellet rewards paired with a predictive cue to compare the temporal dynamics of aversive (90 dB WN; *Figure 2B*, left) and appetitive (reward) conditioning (*Figure 2B*, right). Changes in dopamine are illustrated across time (*Figure 2B*, top), and in order to quantify the shift of the dopamine response from the US to the CS, we calculated the ratio between the areas under the curve (AUC) between them during the very first trials of conditioning ('early day 1'), the rest of the trials of day 1 ('late day 1'), and on the sixth day of conditioning ('day 6') (see *Figure 2B*, bottom). During aversive conditioning, we found, using a mixed-effects analysis, a significant main effect of the degree of conditioning (F(0.9586, 2.396) = 117.3, p=0.0043), and post-hoc testing using Tukey's multiple-comparisons test reveals significant differences between the ratios of day 1 early trials and day 1 later trials (p=0.0138), as well as between day 1 early trials and day 6 (p=0.0318). However, no difference was observed between day 1 later trials and day 6 (p=0.9852). During appetitive conditioning, we also found a main effect of the degree of conditioning on the ratio between the US and the CS (F(1.034,8.788) = 13.88, p=0.0047), and, in contrast to aversive conditioning, the ratio on day 6 is significantly different from both day 1 early trials (p=0.0102) and day 1 later trials (p<0.0001). In addition, we found a significant difference between day 1 early trials and day 1 later trials (p=0.0441). The comparison of conditioned behavioral responses to 90 dB WN between day 1 and day 6 using Wilcoxon signed-rank tests and a Holm–Bonferroni correction for multiple comparisons reveals an increase in locomotion speed (*Figure 2C*), compared

to baseline, which, on day 1, was restricted to the WN epoch alone (cue: Z = –0.876, p=0.381; WN: Z = –3.621, p<0.0001). On day 6, we observe an increase in locomotion speed during both the cue (Z = –3.053, p=0.002) and WN (Z = –3.621, p<0.0001) epoch compared to baseline, which are both also significantly higher compared to day 1 (cue: Z = – 2.485, p=0.013; WN: Z = –2.343, p=0.019). During appetitive conditioning, we see the same temporal evolution in the conditioned response, where rats approach the reward magazine more during the cue epoch on day 6 compared to day 1 (t(9) = 6.962, p<0.0001). However, during appetitive conditioning, but not during aversive conditioning, an almost complete shift of the dopamine response from the CS to the US occurred. Together, these results demonstrate distinct differences in the temporal dynamics of dopamine signaling during aversive and appetitive conditioning.

## Reliability and extinction of cue-induced dopamine signaling

To test the reliability with which WN and its paired cue decrease dopamine concentration across days, we compared behavioral sessions on days 1, 5, and 9. The cue-induced decrease in dopamine concentration became more robust between days 1 and 5, which is explained by the cue acquisition during day 1 (*Figure 2A*). Cue-induced decrease was stable between days 5 and 9, and WN-induced decrease was stable between days 1, 5, and 9 (n = 6, cue: F(1.428, 7.141) = 9.402, p=0.0133; day 1 vs. day 5: p=0.0035; day 1 vs. day 9: p=0.0882; day 5 vs. day 9: 0.7916; WN: F (1.236, 6.178) = 0.03898, p=0.8936; *Figure 2D*). In addition to having monitored the quick acquisition of the cue's dopamine-decreasing properties (*Figure 2A*), to further verify that these properties were a learned response, we tested how fast the association between cue and WN could be extinguished. Rats with well-established cue-WN associations (that were conditioned for more than 6 days) were exposed to 20 consecutive trials in which WN was omitted. Using a Friedman test, we found a significant effect of extinction ($\chi^2$(2) = 8.4, p=0.005). Post-hoc analysis using a Dunn's multiple-comparison test revealed a significant difference between the decrease in extracellular dopamine concentration of trials 1–5 and trials 16–20 (p=0.0133, *Figure 2E*), with the latter no longer showing a decrease in dopamine. Thus, over the course of 15 extinction trials, the cue lost its conditioned dopamine response.

## NAC dopamine does not reflect WN intensity and WN-induced behavior

For the previous experiments, we used WN with an intensity of 90 dB. We asked whether WN of different intensities would differentially influence extracellular dopamine in the NAC since we observed increased avoidance of higher intensities of WN (*Figure 1C*) and greater locomotion in response to higher intensities of WN (*Figure 1D*, right panel). First, we tested whether there is a dose–response relationship between different WN intensities and dopamine. We exposed rats (n = 6) to four different intensities of WN (70, 80, 90, and 96 dB), which were delivered in a semi-random order (*Figure 2F*). Although all WN intensities decreased dopamine release, we found no significant effect of intensity on extracellular dopamine (F(2.380, 11.90) = 0.1655, p=0.8813). In a different experiment, we trained rats (n = 9) on an aversive conditioning paradigm in which two cues (2 kHz or 8 kHz tones) predicted exposure to 6 s of either 80 dB or 96 dB WN, respectively (*Figure 2G*). Again, no significant differences were found in extracellular dopamine release between WN intensities (Z = –0.178, p=0.859), nor between the effects of their respective predicting cues (Z = –0.059, p=0.953). Both of these experiments indicate that extracellular dopamine in the NAC does not encode WN intensity, and, therefore, the relative aversiveness or aversive value of WN is not encoded by NAC dopamine.

Many studies have demonstrated the involvement of dopamine in movement (e.g., *Syed et al., 2016*; *da Silva et al., 2018*; *Coddington and Dudman, 2019*). Therefore, we tested whether a correlation between locomotion speed and extracellular dopamine concentration existed during cue and WN epochs. We performed a trial-by-trial analysis for both 80 dB (*Figure 2H*, top) and 96 dB (*Figure 2H*, bottom) WN exposures and found no correlation during the cue (80 dB: $R^2$ = 0.0004, p=0.7223; 96 dB: $R^2$ = 0.0003, p=0.7360) nor during WN (80 dB: $R^2$ = 0.0016, p=0.4578; 96 dB: $R^2$ = 0.0011, p=0.5374). The dots in the inset graphs represent the $R^2$ values of the average locomotion speed and dopamine concentrations during the cue and WN period in the recording session for each individual animal.

## NAC dopamine signals contain little prediction error

Although the WN-predictive, conditioned cue acquired the ability to reliably suppress NAC dopamine release, no substantial transfer of this effect from US to CS occurred (a prerequisite for a prediction error signal). To further evaluate whether NAC dopamine might function as an APE, we introduced several deviations from the expected outcomes. First, we exposed rats to two different cues (2 kHz or 8 kHz tones) that were associated with different probabilities followed by either 80 dB or 96 dB WN (see *Figure 3A*). Even though dopamine did not encode WN intensity, we hypothesized that dopamine may nonetheless convey an error component to be reflected as diminished dopamine decrease during the better-than-expected condition (occurrence of low-probability [25%] 80 dB WN), and augmented decrease in dopamine during the worse-than-expected condition (occurrence of low-probability [25%] 96 dB WN). As expected, we did not observe differences in dopamine release during the cue epoch due to the uncertainty of which intensity would follow (tone 1: $t(8) = 0.5983$, $p=0.5662$, *Figure 3B*; tone 2: $t(8) = 1.432$, $p=0.1901$, *Figure 3C*). However, during the WN epoch, we also did not find significant differences between the two intensities, neither for tone 1 ($t(8) = 0.6698$, $p=0.5218$), nor for tone 2 ($t(8) = 0.4452$, $p=0.6680$). Together, these results indicate that the prediction error of outcomes deviating from expected probability is not encoded by NAC dopamine concentrations.

Since WN intensity had little influence on dopamine release, we investigated whether an APE was detectable when deviations from the expected outcome occurred in the temporal domain (i.e., duration of WN). We randomly exposed rats to a small number of probe trials with (a) a longer-than-predicted 90 dB WN (12 s instead of 6 s), in other words, a worse-than-predicted outcome (*Figure 3D*); (b) omitted WN, in other words, a better-than-predicted outcome (*Figure 3E*); or (c) unpredicted WN, in other words, another version of a worse-than-predicted outcome, but in this case lacking the prediction completely (*Figure 3F*). These three types of probe trials were implemented in a session in which the first 30 trials consisted of exclusive, deterministic pairings of cue (5 s) and 90 dB WN (6 s), after which these regular predicted WN trials were intermixed with the abovementioned probe trials. During the worse-than-predicted WN trials where the WN was extended by 6 s, we observed an extended suppression of dopamine, which decreased at the same rate as during the initial 6 s, and which ceased immediately upon termination of WN, resulting in a overall lower dopamine concentration in the 11–25 s epoch (*Figure 3D*; $t(10) = 1.863$, $p=0.046$, after Holm–Bonferroni correction for multiple comparisons). During better-than-predicted trials (omitted WN), we found overall higher concentrations of dopamine in the 5–20 s epoch (*Figure 3E*; $t(10) = 3.751$, $p=0.0019$, after Holm–Bonferroni correction). For the unpredicted WN, which constitutes a prediction error since the predictive cue is lacking, we aligned dopamine concentrations for predicted and unpredicted WN at its onset, in order to compare the impact of WN per se. We observed significantly lower dopamine concentrations exclusively in the epoch after the termination of the WN (*Figure 3F*; 11–25 s, $t(10) = 2.453$, $p=0.0170$, after Holm–Bonferroni correction), but not during the WN epoch itself.

Although we did not detect a dopamine error signal during exposure to WN (i.e., deviations from expected WN), we hypothesized that such unexpected events may alter dopamine after WN-offset. Thus, we compared the slope (or rate) of change of dopamine concentration during the recovery epoch (after WN cessation) since using slope allows for integration of the change in dopamine concentration over time, when the animals were presented with deviations from the predicted aversive event. Specifically, we found no significant slope difference between fully predicted WN trials and 'worse-than-expected' trials ($t(10) = 0.1511$, $p=0.4415$, after Holm–Bonferroni correction, *Figure 3D*, bar graph) or 'better-than-expected' trials ($t(10) = 0.4809$, $p=0.3205$, after Holm–Bonferroni correction, *Figure 3E*, bar graph). Thus, when our rats were exposed to unexpectedly extended WN or to the unexpected omission of WN, dopamine concentration reflected only the duration of WN exposure, but not a prediction error. In contrast, we found a significant difference in recovery slope between unexpected and expected WN ($t(10) = 2.895$, $p=0.0080$, after Holm–Bonferroni correction, *Figure 3F*, bar graph), which indicates that, in the case of an unexpected aversive stimulus, dopamine does not only track the duration of this aversive stimulus, but displays a differential response and, thus, may serve as a qualitative teaching signal.

## Dopamine integrates information about appetitive and aversive stimuli

We then investigated whether dopamine could still encode rewards during ongoing WN exposure, that is, while extracellular dopamine concentrations are continuously decreasing. To test this, we delivered

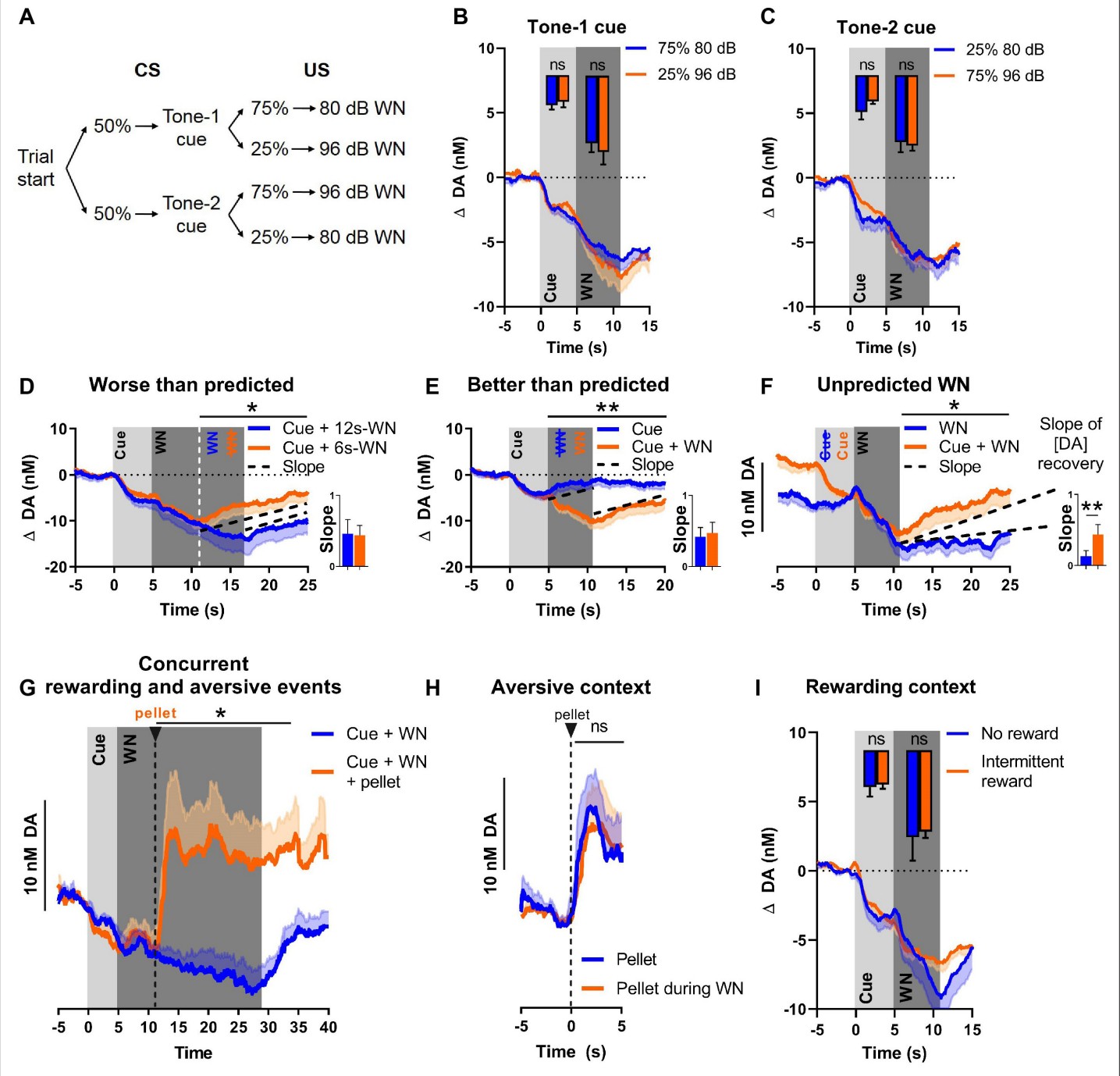

**Figure 3.** Nucleus accumbens core (NAC) dopamine consistently tracks prediction and duration of white noise (WN) with little aversive prediction error function. (**A**) Trial structure of the probabilistic Pavlovian WN task. (**B**) Dopamine concentration in the probabilistic task during the presentation of tone-1 cue, which was followed by 80 dB WN (blue) in 75% of trials and by 96 dB WN (orange) in the remaining 25% of trials. Bar graph inset: no significant differences in average dopamine concentration (n = 9) during cue (t(8) = 0.5983, p=0.5662) and WN (t(8) = 0.6698, p=0.5218). (**C**) Dopamine concentration in the probabilistic task during the presentation of tone-2 cue, which was followed by 80 dB WN (blue) in 25% of trials and by 96 dB WN (orange) in the remaining 75% of trials. Bar graph inset: no significant differences in average dopamine concentration (n = 9) during cue (t(8) = 1.432, p=0.1901) and WN (t(8) = 0.4452, p=0.6680). (**D**) Comparison of dopamine between predicted 6 s WN (orange) and worse-than-predicted 12 s WN (blue) (n = 11) demonstrates significantly lower average dopamine in the epoch between 11 and 25 s during worse-than-predicted 12 s WN (t(10) = 1.863, p=0.046). Bar graph (right): slopes of dopamine concentration trajectories (black dotted lines) show no significant difference between worse-than-predicted and predicted WN (Z = –1.432, p=0.0775). (**E**) Comparison of dopamine between predicted 6 s WN (orange) and better-than-predicted, omitted WN (blue) (n = 11) demonstrates significantly higher average dopamine in the epoch between 5 and 20 s during better-than-

*Figure 3 continued*

predicted, omitted WN (t(10) = 3.751, p=0.0019). Bar graph (right): slopes of dopamine concentration trajectories (black dotted lines) show no significant difference between better-than-predicted and predicted WN (Z = 1.334, p=0.091). (**F**) Comparison of dopamine between predicted 6 s WN (orange) and unpredicted 6 s WN (blue) (n = 11) demonstrates significantly lower average dopamine in the epoch between 11 and 25 s during unpredicted WN (t(10) = 2.453, p=0.0170). Bar graph (right): slopes of dopamine concentration trajectories (black dotted lines) show a significantly flatter slope during unpredicted WN compared to predicted WN (Z = –2.490, p=0.0065). (**G**) Dopamine release during prolonged 24 s WN exposure (blue) continues to incrementally decrease over time. Unexpected reward delivery (n = 6) during such 24 s WN (orange) induces an increase in dopamine in the epoch between 11 and 35 s (t(5) = 3.108, p=0.0266). (**H**) Comparison of dopamine after unexpected pellet delivery (blue) and after unexpected pellet delivery during WN exposure (orange) (n = 6) shows no significant difference in average dopamine in the epoch between 0 and 5 s (t(5) = 0.08753, p=0.9336). (**I**) Comparison of dopamine during WN exposure in a testing context without rewards (blue) and a testing context with intermittent rewards (orange) (n = 6) shows no significant difference in average dopamine during the cue (t(5) = 0.2841, p=0.7877) and WN (t(5) = 0.3151, p=0.7654). Data are mean + SEM or - SEM. *p<0.05, **p<0.01.

food pellets unexpectedly during a prolonged WN epoch. We observed a significant increase in dopamine release upon pellet delivery (t(5) = 3.108, p=0.0266, *Figure 3G*), which was comparable to the increase in dopamine release we observed upon pellet delivery in the absence of WN (t(5) = 0.08753, p=0.9336, *Figure 3H*). These results indicate that dopamine is still responsive to rewarding events during an aversive event and, thus, integrates information about appetitive and aversive events.

A previous study reported that dopamine is more prone to encode an APE in an experimental context with a low probability of intermittent reward delivery (*Matsumoto et al., 2016*). Thus, we compared dopamine release during cue and WN exposure embedded in two task contexts with different reward probabilities (i.e., different 'reward contexts'). In the first task context, no rewards were delivered during the entire session, whereas in the second context a low chance of reward delivery existed (reward trial probability = 0.1). We did not observe a significant difference in dopamine concentration between these reward contexts during the cue epoch (t(5) = 0.2841, p=0.7877), nor during the WN epoch (t(5) = 0.3151, p=0.7654) (*Figure 3I*). Consistently, in another experiment, we did not observe a significant difference in dopamine concentration (during cue and WN epochs) when comparing a no-reward context with a high-reward context (reward trial probability = 0.5; cue: t(17) = 1.448, p=0.0829; WN: t(17) = 1.428, p=0.0857; data not shown).

## Discussion

In this study, we set out to delineate the role of the dopamine system in processing aversive stimuli, by systematically investigating subsecond fluctuations in rat NAC dopamine concentration in response to an aversive auditory stimulus (WN), as well as its prediction by auditory tone cues. First, we demonstrated that WN-induced behavioral activation is WN-intensity-dependent and validated the aversiveness of WN in real-time place aversion, approach-avoidance foraging, and operant tasks, where we found that WN aversiveness scales with WN intensity. Trial-by-trial analysis of the first WN exposures revealed that WN as an unconditioned stimulus diminishes the concentration of extracellular dopamine in the NAC, and that a predicting cue rapidly takes on the role of conditioned stimulus (reversible by extinction), eliciting WN-like behavioral activation and dopamine depression. Dopamine during cue and WN was not correlated with locomotion speed. In contrast to appetitive conditioning, only a very limited temporal shift of the dopamine response from WN to the cue occurred. Dopamine responses to WN and its predictive cue were not affected by aversive value (WN intensity), context valence (introduction of intermittent rewards), or probabilistic contingencies. Instead, prediction and duration of the aversive WN were accompanied by a relatively slow and steady decrease in NAC dopamine concentration (a declining ramp that continued without plateauing), which was followed by an equally slow recovery of dopamine upon cessation of WN. The slope of this rebounding dopamine ramp was altered only by unpredicted presentation of WN (not by better-than-predicted or worse-than-predicted outcomes), revealing a function of dopamine that sometimes goes beyond simple real-time tracking the presence of conditioned and unconditioned aversive stimuli. Finally, we found the integration of rewarding and aversive stimuli is of parallel nature as WN-associated dopamine depression did not modify the rapid surge of dopamine triggered by unexpected reward delivery. Together, our findings indicate that negative dopamine signals in the NAC mostly track the prediction and duration of aversive events, with few aspects that are consistent with an APE.

## WN is a versatile aversive stimulus that suppresses dopamine release and increases locomotion

We chose WN as an aversive stimulus to probe the limbic dopamine system's role in aversive conditioning as it possesses several merits. First, at intensities that are not prone to cause loss of hearing (*Escabi et al., 2019*), WN is moderately aversive (*Campbell and Bloom, 1965*; *Hughes and Bardo, 1981*), and as such does not induce freezing, but instead provokes mild behavioral activation. This is particularly relevant with regard to studies relating dopamine function to behavioral read-outs since lack of movement is often associated with diminished activity of the dopamine system and, thus, may confound the interpretation of negative dopamine signals in the context of aversive events. Second, WN is reliably effective and tolerated across many trials and sessions, supporting the detection of neuronal signals by providing sufficient data for averaging across trials and enabling complex experiments with varying valence and contingencies. Third, WN is distinct, well-controllable, and easy to produce, where intensity and duration can be titrated effortlessly. Fourth, WN does not require attention to be detected (i.e., the animal will hear it anywhere in an experimental environment). Fifth, animals cannot interfere with WN delivery, as opposed to air puffs or electric foot shocks, which can be influenced by the animal's actions and position (i.e., closing its eyelids or decreasing contact surface with the charged grid floor). Sixth, WN does not interfere with data recording in FSCV, electrophysiology, or fluorescence imaging. Together, the abovementioned merits make WN an experimentally valuable stimulus with great potential to uncover aversion-relevant brain mechanisms.

In this work, we report that WN diminishes extracellular dopamine concentration in the NAC upon first exposure, characteristic of an unconditioned stimulus or primary reinforcer. A predicting cue quickly adopted this property upon subsequent exposures, which was reversible by extinction. Such dopamine responses were stable across trials and sessions. Interestingly, the use of WN revealed a rare relationship between dopamine and behavior: increased locomotion speed was associated with a decrease in dopamine release. Behavioral activation is usually associated with increased dopamine signaling (*Boureau and Dayan, 2011*; *Berridge and Robinson, 1998*; *da Silva et al., 2018*; *Coddington and Dudman, 2019*), whereas a lack of movement or even freezing, depending on stimulus intensity, is often associated with decreased dopamine (e.g., *Oleson et al., 2012*; *Badrinarayan et al., 2012*). These frequently observed association patterns have prompted the hypothesis that the directionality of changes in dopamine concentration reflects the chosen behavioral strategy when confronted with aversive stimuli: an active or a passive reaction (*Badrinarayan et al., 2012*). Our results, however, prove that this hypothesis is not universally applicable. In this context, it would be interesting to assess dopamine responses to greater WN intensities than the ones used here. Such intensities could offer further insight into the relationship between behavior and dopamine dynamics because they are known to elicit freezing (*Rescorla, 1973*; *Ledgerwood et al., 2005*; *Furlong et al., 2016*), albeit with a higher risk of hearing loss (*Escabi et al., 2019*). Freezing elicited via electric shocks is accompanied by decreased NAC dopamine activity (*Badrinarayan et al., 2012*; *Oleson et al., 2012*; *de Jong et al., 2019*; *Stelly et al., 2019*). The behavioral activation we observe in response to WN might reflect an increased motivation to escape, which we cannot ascertain as our task was Pavlovian (thus, without an active avoidance component: the WN was inescapable). Notably, the observed decline in dopamine was not at all correlated with movement on a trial-by-trial basis; thus, it is conceivable that during mild WN exposure, NAC dopamine was uncoupled from its usual, more direct behavioral impact.

## What is and what is not encoded by NAC dopamine?

Many studies have investigated the role of dopamine in aversion by testing the system's reaction to the exposure to aversive stimuli (see above), but only a few scrutinize dopamine's precise function therein, or whether dopamine encodes a 'true' APE. Their conclusions range from 'dopamine is insensitive to aversiveness' (*Fiorillo, 2013*), to the other extreme of 'dopamine serves as an APE' (*Matsumoto et al., 2016*). *Fiorillo, 2013* ruled out the existence of a dopamine APE because (1) dopamine-neuron firing did not differ between presentation of aversive and neutral stimuli, (2) prediction of an aversive event did not affect firing, and (3) no integration of rewarding and aversive values was observed. In contrast, *Matsumoto et al., 2016* found evidence for all three of these requirements and, therefore, concluded that dopamine neurons are capable of encoding a value prediction error (equally for both rewards and aversive stimuli). This discrepancy could partially be explained by the

fact that *Matsumoto et al., 2016* recorded from dopaminergic neurons in the VTA (of mice), whereas the majority of the neurons that *Fiorillo, 2013* recorded were in the substantia nigra (of monkeys). Since we measured extracellular concentrations of NAC dopamine, which is released from terminals that originate from neurons in the VTA (*Ikemoto, 2007*), we expected to find an APE in our data.

Indeed, our results meet Fiorillo's (2013) three requirements for an APE stated above: (1) during the first pairing of cue and WN, when the predictive auditory cue was still neutral, dopamine concentration during the WN epoch differed significantly from that during baseline and cue presentation, but the latter (cue and baseline dopamine) did not differ from each other. (2) After only four cue-WN pairings, cue presentation diminished dopamine concentration, thus prediction of the aversive event did alter dopamine activity. Although we did not find a significant difference in overall dopamine concentration between predicted and unpredicted WN, we did observe a difference in their post-WN recovery slopes. (3) Finally, although we did not detect an 'interactive' integration (modulated signal) of aversive and reward values during concurrent presentation of WN and a food pellet (as the absolute magnitude of released dopamine was equal to that of a pellet delivered outside of WN exposure), both the rewarding and aversive stimuli were encoded in parallel. The signals are thus integrated in the sense that both are processed at the same time, in an additive manner; as opposed to an exclusive organization, where the dopamine system may be 'turned off' or unresponsive towards rewards during the presence of an aversive stimulus. Taken together, up to this point, our results fit best with the conclusion of *Matsumoto et al., 2016*; although a noteworthy contrast is that in our data, context was irrelevant to the magnitude of dopamine response to the aversive event and reward: it made no difference for the acute dopamine response magnitude whether the aversive stimulus was delivered in rewarding contexts or not.

Next, however, we took inspiration from *Hart et al., 2014*, who used a mathematical approach developed by *Caplin and Dean, 2007* to confirm the encoding of RPE signals by NAC dopamine. They used a deterministic and a probabilistic choice task in order to determine whether dopamine signals fulfilled three axioms that were considered necessary for a RPE signal: 'consistent prize ordering,' 'consistent lottery ordering,' and 'no surprise equivalence.' We employed the abovementioned deterministic and probabilistic Pavlovian conditioning tasks to identify an APE, instead of an RPE, by exposing rats to low- and high-dB WN, predicted by two different tones, with either a 100% probability (deterministic task) or with different probabilities (probabilistic task). We did not observe differences in dopamine concentration during the WN epoch in the deterministic task, which fulfills the third axiom (no surprise equivalence), since the prediction error is zero for both of these conditions. But the first two axioms were not fulfilled since we did not detect differences in dopamine during the WN epoch, when different WN intensities were presented with different probabilities. Rats avoided higher-dB WN more than lower-dB WN (*Figure 1B*) and exhibited WN dB-dependent locomotor activation (*Figure 2F*), indicating that WN aversiveness scales with WN intensity and that rats are able to discriminate between different WN intensities. Thus, we conclude that the NAC dopamine signals we observed did not fulfill the axiomatic criteria of an APE, when aversive stimulus intensity or value was varied.

Finally, we performed experiments to probe the dopamine signal in conditions where the aversive stimulus deviated from the expected duration; in other words, when trials were worse or better than predicted based on WN duration, but with a stable intensity of 90 dB. First, we extended WN duration or omitted WN in occasional trials. Extended WN elicited a continuation of the same declining dopamine concentration slope, which ceased promptly at WN cessation, after which dopamine slowly ramped backup toward baseline with a reversed, inclining slope. Thus, although the signal reflected the duration of extended WN, no discernible error component was evident. When WN was omitted, we did not observe an error signal either: instead, again, the signal slowly returned to baseline levels. Second, in another version of the 'worse-than-expected' condition, we occasionally delivered WN unexpectedly, without a preceding cue (after animals had learned the cue-WN association well), and observed a difference in the recovery slope after WN-offset compared to predicted WN. This flattened recovery slope indicates that NAC dopamine signals more than simply track the presence of aversive stimuli; in addition, it may relate to the failed anticipation of an aversive event (based on reliance on the predictive cue), and, thus, indicate altered cue-WN contingencies. In summary, we conclude that dopamine precisely tracks aversive stimulus duration, and the only evidence of an APE-like signal in our data was found after unpredicted WN, whereas several of our other experimental

accounts are incompatible with an APE function of NAC dopamine. This places our results firmly in the middle ground between the no-APE (*Fiorillo, 2013*) and the full-APE conclusions (*Matsumoto et al., 2016*) described above.

## Aversion versus reward

Consistent with most literature (*Badrinarayan et al., 2012*; *Oleson et al., 2012*; *de Jong et al., 2019*; *Stelly et al., 2019*), we found that NAC dopamine encodes rewarding and aversive events with opposite directionality. Furthermore, we report that a cue predicting an aversive stimulus can adopt the ability to prompt dopamine changes the way the aversive stimulus itself would. Taken together, this suggests that NAC dopamine encodes both reward and aversive prediction. However, decreases in dopamine concentration did not scale with WN intensity, unlike what is well-established for reward processing, where reward size or probability is encoded both for the reward itself and for predictive stimuli (*Gan et al., 2010*; *Tobler et al., 2005*; *Watabe-Uchida et al., 2017*). Thus, one could speculate that the diminishing effect of WN on NAC dopamine may be related to pre-attentive processes involved in saliency and novelty – dissociable from NAC value signals (*Redgrave and Gurney, 2006*; *Kutlu et al., 2021*; *Kutlu et al., 2022*). Furthermore, encoding of a prediction error, which is one of the best-characterized features of reward-related dopamine signaling, did not occur for aversive events. Thus, NAC dopamine does not encode aversive and appetitive stimuli (and their prediction) in the same way. Moreover, the basic nature of aversion-related dopamine signals in our data was different from that of rewards. For example, the temporal signal shift toward the earliest predictor of the respective reinforcing stimulus, as described for rewards, is incomplete for aversive conditioning. Another example is that reward-related changes in extracellular dopamine concentration are substantially larger and faster compared to aversive events. These discrepancies may be partially attributable to general differences between dopamine release into and removal from the extracellular space. More specifically, the dopamine system presumably has a bigger dynamic range for increasing activity; it can do so, for example, by increasing the number of cells firing and their firing frequency (and thereby the total number of dopamine-containing vesicles being released). In contrast, dopamine-signaling reduction cannot drop below a certain point since the cells' maximum response is to cease firing altogether and extracellular dopamine can only be removed relatively slowly or must diffuse away. This disparity could translate into a structurally limiting factor on what can be encoded by a reduction in dopamine concentration and explain some of the abovementioned differences in function. However, the slow-ramping declining and recovery slopes we observed do not reflect the system limits since the very first exposure to WN resulted in a steeper decline and rewards given during WN resulted in steeper increases. Furthermore, disparate qualitative differences were also found in NAC-dopamine responses to the presentation of ultrasonic vocalizations that are associated with rewarding and aversive events (*Willuhn et al., 2014b*). Taken together, our results indicate that there are a few similarities between dopamine encoding of rewards and aversive stimuli, but overall we found more differences between them – hinting at aversive events being encoded by NAC dopamine more rudimentarily in a qualitative instead of quantitative fashion.

In summary, our findings demonstrate that WN is a valuable and versatile aversive stimulus that is well-suited to probe how the brain processes aversive stimuli. Overall, we conclude that dopamine tracks the anticipation and duration of an aversive event. This tracking materializes as a perpetually declining dopamine ramp that progresses without altering its slope until offset of the aversive stimulus (even WN lasting for 24 s did not reach a plateau of minimal dopamine concentration). Such aversion tracking may play an anticipatory role for certain defensive behaviors since the animals were behaviorally activated during the aversive event. Furthermore, we speculate that these slowly ramping aversion signals may contribute to a qualitative learning signal (other than a quantitative or scalar APE signal) since the unexpected aversive stimulus elicited a response beyond simply tracking the stimulus. Thus, we conclude that dopamine tracks both positive and negative valence in their temporal aspects and prediction, but that quantitatively speaking, the exact value and error is only encoded for rewards, in the upward direction of NAC dopamine concentration. This implies that aversive value and APEs are encoded in other brain regions.

# Materials and methods

## Animals

Adult male Long–Evans rats (300–400 g; Janvier Labs, France) were housed individually and kept on a reversed light–dark cycle (light on from 20:00 till 8:00) with controlled temperature and humidity. All animal procedures were in accordance with the Dutch and European laws and approved by the Animal Experimentation Committee of the Royal Netherlands Academy of Arts and Sciences under CCD license numbers AVD801002015126 and AVD80100202014245. In total, 37 rats underwent surgery, 21 of which exhibited a functional FSCV electrode with a histologically verified location in the NAC, and were therefore included in the study. An additional 64 rats, which did not undergo surgery, were used for behavioral tasks (*Source data 1*). All rats were food-restricted to 90% of their free-feeding bodyweight, and water was provided ad libitum.

## Stereotaxic surgery

Rats were induced under isoflurane anesthesia and placed into the stereotaxic frame on an isothermal pad maintaining body temperature. The analgesic Metacam (0.2 mg meloxicam/100 g) was injected subcutaneously and the shaved scalp was disinfected using 70% ethanol. Upon incision of the scalp, it was treated with lidocaine (100 mg/ml). Holes were drilled in the cranium and the dura mater was cleared for targeting the NAC (1.2 mm AP, 1.5 mm ML, and –7.1 DV; *Paxinos and Watson, 2007*). Chronic carbon-fiber electrodes (*Clark et al., 2010*), made in-house, were positioned in the NAC, and an Ag/AgCl reference electrode was placed in a separate part of the forebrain. The electrodes were secured to screws in the skull using cranioplastic cement. Following surgery, rats received subcutaneous injection of 2 ml saline and were placed in a temperature-controlled cabinet to be monitored for an hour. Rats were given 1–2 weeks post-surgery to recover before food restriction, behavioral training, and recording.

## Behavioral procedures

All behavioral experiments, except the place-aversion and foraging tasks, were conducted in modified operant boxes (32 × 30 × 29 cm, Med Associates Inc), equipped with a food magazine (connected to an automated food-pellet dispenser) flanked by two retractable levers (with cue lights), a house light, multiple tone generators, two WN generators, and metal grid floors (Med Associates Inc). Each operant box was surveilled by a video camera. The boxes were housed in metal Faraday cages that were insulated with sound-absorbing polyurethane foam.

### Real-time place aversion

Rats (n = 10) were placed for 30 min in a light-shielded, square, Perspex open field (60 × 60 × 60 cm), made in-house (Netherlands Institute for Neuroscience [NIN] mechanical workshop). A camera mounted in the center above the open field recorded the position of the rat, which was tracked in real time by the open-source software Bonsai (*Lopes et al., 2015*). One quadrant of the open field was paired with exposure to 90 dB WN, which was produced by a WN generator from Med Associates Inc, mounted on top of one of the open-field walls. WN was automatically switched on as long as the head of the rat was present in the chosen quadrant, and switched off as soon as the rat exited the quadrant. The WN-quadrant position was fixed throughout the session.

In the first experiment, the WN-quadrant position was assigned randomly for each rat (*Figure 1A*). The percentage of time rats spent in the WN-paired quadrant was compared to chance level (25%). In the second experiment, another cohort of rats was placed into the open field without WN exposure to assess their preferred quadrant (*Figure 1B*). After this session, rats were placed in the open field in a second session, where now the entry into their preferred quadrant led to 90 dB WN exposure (*Figure 1B*). The percentage of time rats spent in the preferred quadrant was compared to the time spent in the WN-paired quadrant.

### Approach-avoidance foraging task

A cohort of rats (n = 12) performed in an open-field foraging task (*Figure 1C*). For this purpose, a moveable grid floor that covered one quadrant of the open field was installed. Ten food pellets were placed between the grid-floor bars before the rats were placed in the middle of the open field for

each 10 min session. After 1 day of habituation to the setup, rats underwent a series of sessions, where they were exposed to 0, 70, 80, 90, or 96 dB of WN, or a 70 dB 3 kHz tone upon entrance to the grid-floor quadrant, where they could forage for the pellets by reaching between the grid bars. Rats were exposed to each auditory stimulus twice during foraging in two rounds of sessions, where rats underwent exposure to all WN intensities and the tone in a first session, before exposure to each in a second session. The sequence of sessions proceeded from lowest to highest WN intensity, followed by the tone session. The 3 kHz tone was administered to evaluate whether WN was more aversive than other auditory stimuli. The location of the grid-floor quadrant was changed between rounds.

## WN and reward choice task

Rats (n = 9) were trained to press one of the two levers in the operant box to receive food-pellet rewards (Dustless Precision Pellets, 45 mg, Bio-Serv). During the first training days, a single lever was inserted at variable intertrial intervals. Pressing this lever prompted delivery of a single food pellet and immediate retraction of the lever. Omissions (no lever press for 10 s) resulted in 10 s house-light illumination. After reaching a 90% success rate, the other lever was introduced in training sessions that consisted of 20 'forced' trials, in which one of the two levers was presented, followed by 80 choice trials where both levers were presented. Any lever press resulted in delivery of a single food pellet and retraction of extended levers (marking the end of the trial). After five consecutive sessions with over 90% success rate, we paired reward delivery of one of the levers with simultaneous 5 s of 90 dB WN exposure. Rats were trained under these contingencies for 4 days, after which half of the animals (n = 5) were switched to 96 dB and the other half to 70 dB WN (n = 4). After four sessions, WN intensities were transposed between the two groups of animals for an additional four sessions, so that every animal received each WN intensity. In each of these WN sessions, animals could earn a maximum of 100 pellets (in 100 trials). Both levers were presented simultaneously at a variable intertrial-interval averaging 25 s (range: 15–35 s). Just as at the start of training, one lever press induced immediate retraction of both levers and prompted reward delivery (end of trial), whereas omissions (no press within 10 s upon lever insertion) ended the trial and resulted in 10 s house-light illumination. We compared the relative number of WN-paired lever presses across different WN intensities.

## FSCV during aversive Pavlovian conditioning with 90 dB WN

On the first day of aversive Pavlovian conditioning, a new group of 16 rats was tethered to the FSCV recording equipment and placed in the operant box. In this and all paradigms described below, prior to behavioral session start, two unexpected deliveries of a single food pellet (spaced apart by 2 min) confirmed electrode viability to detect dopamine. The session started with the illumination of the house light. The first 30 trials consisted of the presentation of a 5 s cue (1.5 kHz, 75 dB tone) followed by 6 s of WN (90 dB). Trials were separated by a variable intertrial interval of 60 s (range: 30–90 s). For a subset of the rats (n = 11), these initial 30 trials were followed by 55 trials, in which 5 s cue/6 s WN pairings were randomly mixed with four trials with unpredicted WN (6 s of 90 dB WN without cue), four trials with WN omission (5 s cue without WN), and four trials with 5 s cue followed by 12 s of 90 dB WN (longer-than-expected condition).

A subset of the initial 16 rats (n = 4) was conditioned for an additional 5 days (days 2–6), of which the first 4 days consisted of sessions with 30 trials of pairings of 5 s cue/6 s with 90 dB WN, and on the fifth day (sixth day of conditioning in total) another FSCV recording session took place (as described for day 1). An additional group of animals (n = 13) without implanted FSCV electrodes were conditioned for 6 days in order to characterize behavioral responses to different WN intensities.

For the analysis of the first 30 conditioning trials, we compared the average dopamine concentration during the cue (5 s) and the WN (6 s) epoch to baseline (−5 to 0 s before cue onset). To analyze trials with different contingencies (unpredicted, omitted, or longer WN), we compared average dopamine during the relevant epochs (unpredicted WN: 11–25 s after cue onset; better than predicted [omitted WN]: 5–20 s after cue onset; worse than predicted [longer WN]: 11–25 s after cue onset) with average dopamine in the respective epochs in immediately preceding trials 5 s cue/6 s WN pairings (trials 25–30), during which dopamine decreases had stabilized and were unaffected by different contingencies. Slopes of dopamine traces were compared between trials with different contingencies and predicted WN trials since using the slope allows for integration of the change in dopamine

concentration over time as opposed to averaging concentrations over an epoch (in which there is no integration over time). All traces were aligned before WN onset.

To compare dopamine concentration during cue and WN between days 1 and 6 and to quantify the shift of dopamine release from the US to the CS, we subdivided the results of day 1 into 'day 1 (early)' (trials 2–4; trial 1 was excluded to remove the saliency response to the first cue exposure) and 'day 1 (late)' (trials 5–30). We calculated the ratio between US and CS dopamine signals as a deviation from baseline (in the respective up or down direction). For aversive conditioning, this ratio was determined by (area above the curve of the WN epoch)/(area above the curve of the cue epoch). For appetitive conditioning, this ratio was determined by (area under the curve of the pellet epoch)/(area under the curve of the cue epoch).

## Appetitive Pavlovian conditioning

Rats (n = 10) were placed in the operant box, and on days 1 and 6, they were tethered to the FSCV recording equipment. Illumination of the house light signaled the beginning of the session. Sessions consisted of 40 pairings of cue-light illumination (5 s) with a pellet delivery (delivered immediately after cue offset), which were separated by variable intertrial intervals averaging 60 s (range: 30–90 s).

## WN dose response

Rats (n = 6) were tethered to the FSCV recording equipment and placed into the operant box. The two WN generators with custom-made volume control dials (NIN mechanical workshop) were used to switch between different WN intensities. The FSCV recording session consisted of six blocks in which two different WN intensities (70, 80, 90, or 96 dB) were presented for 6 s in random order, four times each, with a variable intertrial-interval averaging 30 s (range: 25–35 s). Between blocks, the volume dial was used to change WN intensities. During the different blocks, all WN intensities were presented in pairs of two and, therefore, each intensity was presented in 3 of the 6 blocks and played 12 times in total. We compared the average dopamine concentration during the WN exposures between the different intensities.

An additional group of rats (n = 13) without implanted FSCV electrodes were placed into an operant box and underwent WN exposure in order to characterize behavioral responses to the four different WN intensities (70, 80, 90, or 96 dB; randomly ordered in blocks of 15 trials) presented for a duration of 6 s per trial, followed by an average variable intertrial interval of 60 s (range: 30–90 s). Before the start of each block, three food pellets were delivered with a variable intertrial interval averaging 30 s (range: 20–40). We compared the average baseline-subtracted locomotion speed of the animals during the WN exposures between the different intensities.

## Aversive Pavlovian conditioning with 80 dB and 96 dB WN

Rats (n = 9) underwent four aversive conditioning sessions in the operant box in which a 2 kHz and 8 kHz tone (5 s cue) predicted the exposure to 80 dB or 96 dB WN (6 s), respectively. Sessions consisted of 88 trials, of which 40 trials with 80 dB WN and 40 trials with 96 dB WN, predicted by their respective cues, were presented in random order. In the remaining eight trials, an unpredicted food pellet was delivered. These deliveries were distributed across the session so that in every block of 10 WN exposures, one pellet was delivered at a random trial number. Trials were separated by variable intertrial-intervals averaging 60 s (range: 30–90 s). On the fourth conditioning day, a recording session took place, for which the rats were connected to the FSCV recording equipment. We compared average dopamine concentrations during the cue (5 s) and during the WN (6 s) epochs.

During the subsequent four aversive conditioning sessions, we changed the probability of exposure to 80 dB and 96 dB WN following their associated cues. The total number of presentations of tone 1, tone 2, WN, and pellet deliveries remained the same. However, tone 1 was now followed by 80 dB WN (6 s) in 75% of the trials and by 96 dB WN (6 s) during the remaining 25% of the trials. Tone 2 was followed by 96 dB WN (6 s) in 75% of the trials and 80 dB WN (6 s) during 25% of the trials. A recording session took place on the fourth conditioning day. We compared the average dopamine concentrations during the cue (5 s) and WN (6 s) epochs.

### Concurrent reward and WN, and cue extinction

Rats (n = 6) were connected to the recording set up and placed in the operant box. The conditioning session began with 10 pairings of the 5 s cue (1.5 kHz tone) and 6 s WN (90 dB). Next followed a block of 20 trials pairing the 5 s cue and 24 s of WN; during half of these trials (randomized), a pellet was delivered 6 s into the WN exposure. The recording session was concluded with a block of 20 extinction trials, in which only the 5 s cue was delivered. This recording session was the last to take place, the rats had experienced 9–11 conditioning sessions prior to this recording.

## FSCV measurements and analysis

As described previously (*Willuhn et al., 2014a*), FSCV was used to detect subsecond changes in extracellular concentration of dopamine using chronically implanted carbon-fiber microsensors that were connected to a head-mounted voltammetric amplifier, interfaced with a PC-driven data-acquisition and analysis system (National Instruments) through an electrical commutator (Crist), which was mounted above the test chamber. Every 100 ms, voltammetric scans were repeated to achieve a sampling rate of 10 Hz. The electrical potential of the carbon-fiber electrode was linearly ramped from –0.4 V versus Ag/AgCl to +1.3 V (anodic sweep) and back (cathodic sweep) at 400 V/s (8.5 ms total scan time) during each voltammetric scan, and held at –0.4 V between scans. Dopamine is oxidized during the anodic sweep, if present at the surface of the electrode, forming dopamine-o-quinone (peak reaction detected around +0.7 V), which is reduced back to dopamine in the cathodic sweep (peak reaction detected around –0.3 V). The ensuing flux of electrons is measured as current and is directly proportional to the number of molecules that undergo electrolysis. The background-subtracted, time-resolved current obtained from each scan provides a chemical signature characteristic of the analyte, allowing resolution of dopamine from other substances (*Phillips and Wightman, 2003*). Chemometric analysis with a standard training set was used to isolate dopamine from the voltammetric signal (*Clark et al., 2010*). All data was smoothed with a moving 10-point median filter and baseline (set at 1 s before cue onset or, in case of an absent cue, 1 s before WN onset) subtraction was performed on a trial-by-trial basis prior to analysis of average concentration. Analyses were performed on dopamine concentration during cue (5 s) and WN (6 s) epochs and were compared to baseline dopamine concentrations or to the same epoch in a different experimental condition. Prior to each FSCV recording session, two unexpected deliveries of a single food pellet (spaced apart by 2 min) confirmed electrode viability to detect dopamine. Animals were excluded from analysis when (1) a lack of dopamine release in response to unexpected pellets before start of the behavioral session, and (2) FSCV recording amplitude background noise that was larger than 1nA in amplitude.

## Analysis of operant box behavior

DeepLabCut software (*Mathis et al., 2018*) was used to track rat movement in the operant box using video data recorded during FSCV measurements. This tracking data was analyzed in MATLAB (The MathWorks, Inc, version 2019a) to determine distance to the reward magazine and speed of movement (cm/s). Analyses were performed using the average distance or locomotion speed during the cue (5 s) or WN (6 s) epochs. During the WN- and reward-choice task, the number of presses on each lever was registered via an automated procedure.

## Histological verification of recording sites

After completion of the experiments, rats were deeply anesthetized using a lethal dose of pentobarbital. Recording sites were marked with an electrolytic lesion before transcardial perfusion with saline, followed by 4% paraformaldehyde (PFA). Brains were removed and post-fixed in PFA for 24 hr after which they were placed in 30% sucrose for cryoprotection. The brains were rapidly frozen using an isopentane bath, sliced on a cryostat (50 μm coronal sections, –20°C), and stained with cresyl violet.

## Statistical analysis

FSCV and behavioral data were analyzed using one- or two-tailed paired or unpaired *t*-tests, repeated-measures ANOVAs, regression analysis, or their nonparametric equivalents when appropriate. Post-hoc analyses were conducted when necessary and p-values were adjusted when multiple comparisons were made. Statistical analyses were performed using Prism (GraphPad Software) and SPSS statistics version 25.0 (IBM); graphical representations were made using Prism. Statistical significance was set

to p<0.05. Sample size was not explicitly determined by a power analysis when the study was being designed, but was, instead, based on the lab's experience with this type of data.

## Acknowledgements

We thank Ralph Hamelink and Nicole Yee for their technical support, Matthijs Feenstra for his input on the manuscript, Lucia Economico for illustrations, and Linda Dekker for histology.

## Additional information

### Funding

| Funder | Grant reference number | Author |
|---|---|---|
| European Research Council | ERC-2014-STG 638013 | Ingo Willuhn |
| Nederlandse Organisatie voor Wetenschappelijk Onderzoek | VIDI 864.14.010, 2015/06367/ALW | Ingo Willuhn |
| Nederlandse Organisatie voor Wetenschappelijk Onderzoek | BRAINSCAPES 024.004.012, Gravitation program | Ingo Willuhn |

The funders had no role in study design, data collection and interpretation, or the decision to submit the work for publication.

### Author contributions

Jessica N Goedhoop, Conceptualization, Data curation, Formal analysis, Validation, Investigation, Methodology, Writing – original draft, Writing – review and editing; Bastijn JG van den Boom, Data curation, Formal analysis, Validation, Investigation, Visualization; Rhiannon Robke, Data curation, Formal analysis, Investigation, Visualization; Felice Veen, Data curation, Formal analysis, Investigation; Lizz Fellinger, Data curation, Investigation; Wouter van Elzelingen, Investigation, Methodology; Tara Arbab, Conceptualization, Visualization, Writing – original draft, Writing – review and editing; Ingo Willuhn, Conceptualization, Formal analysis, Supervision, Funding acquisition, Visualization, Methodology, Writing – original draft, Project administration, Writing – review and editing

### Author ORCIDs

Jessica N Goedhoop http://orcid.org/0000-0001-5077-5697
Bastijn JG van den Boom http://orcid.org/0000-0002-0853-3763
Tara Arbab http://orcid.org/0000-0002-7294-7223
Ingo Willuhn http://orcid.org/0000-0001-6540-6894

### Ethics

All animal procedures were in accordance with the Dutch and European laws and approved by the Animal Experimentation Committee of the Royal Netherlands Academy of Arts and Sciences. CCD license numbers AVD801002015126 and AVD80100202014245.

### Decision letter and Author response

Decision letter https://doi.org/10.7554/eLife.82711.sa1
Author response https://doi.org/10.7554/eLife.82711.sa2

## Additional files

### Supplementary files

- Transparent reporting form
- MDAR checklist
- Source data 1. Experiment overview.

## Data availability

Data, statistics, and code at https://osf.io/8p37x/.

The following dataset was generated:

| Author(s) | Year | Dataset title | Dataset URL | Database and Identifier |
|---|---|---|---|---|
| Goedhoop J, Willuhn I | 2021 | Nucleus-accumbens dopamine tracks aversive stimulus duration and prediction but not value or prediction error | https://osf.io/8p37x/ | Open Science Framework, 8p37x |

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
