## [Editor Report]

The article by Goedhoop et al. provides an important analysis of the role of terminal dopamine release in the nucleus accumbens in processing aversive events that will be of value to researchers interested in the neural mechanisms of reinforcement learning and computational modeling of dopamine function. Using a variety of conditions, the authors provide convincing data in support of the role of accumbal dopamine release in processing aversive events that situate the current report among growing interest and mounting investigations into the role of dopamine in aversion.

---

## [Decision Letter]

**Decision letter after peer review:**

[Editors’ note: the authors submitted for reconsideration following the decision after peer review. What follows is the decision letter after the first round of review.]

Thank you for submitting your work entitled "Nucleus-accumbens dopamine tracks aversive-stimulus duration and prediction but not value or prediction error" for consideration by *eLife*. Your article has been reviewed by 3 peer reviewers, one of whom is a member of our Board of Reviewing Editors, and the evaluation has been overseen by a Senior Editor. The following individuals involved in review of your submission have agreed to reveal their identity: Erik Oleson (Reviewer #3).

Comments to the Authors:

We are sorry to say that, after consultation with the reviewers, we have decided that your work will not be considered further for publication by *eLife*.

There was overwhelming enthusiasm for the manuscript and agreement regarding the strengths of the paper by the reviewers. These include the timely and important question of the role of dopamine in aversion, as well as the experimental approach especially the variety of behavioural tasks used. The strengths have been expressed by the reviewers in their individual comments (see below). However, a number of concerns were raised that require substantial reframing (punishment vs. aversive stimulus) of the manuscript as well as additional data collection. Although the reviewers' individual comments are appended to the decision letter, I wanted to give a brief overview of the discussion that led to the decision. One of the key aspects of the paper is the idea that dopamine concentration should be modulated in a white noise intensity-specific manner. However, the behavioural evidence that the white could be discriminated at different intensities came using different parameters compared to the neurochemical experiment. It is paramount to show that given these parameters the animals can indeed discriminate the white intensities as this is key to the argument of whether dopamine tracks aversive prediction error. Relatedly, behavioural evidence of conditioning is needed in the context of the recordings. The influence of baseline choice and amplitude vs. duration of the signal on the data need to be explored further. A more specific concern regarding the lack of correspondence between Figure 2B trace and bar graphs. It was unclear to the reviewers how the trace data correspond to the bar graphs below the trace. The statistical methods and degrees of freedom were also unclear. While this is a short overview of some of the concerns that were flagged, it is not exhaustive, further concerns are noted in the individual reviews. The reviewers and editors felt that addressing these concerns is beyond the scope of a standard *eLife* revision. However, given the enthusiasm for the manuscript if you are able to well address each of the reviewers concerns, we would be willing reconsider the manuscript for publication at *eLife* as a new submission, with a point by point rebuttal to each concern.

*Reviewer #1 (Recommendations for the authors):*

Understanding the role of dopamine beyond reward and specifically in aversion is of fundamental importance. The paper set out to examine whether dopamine concentration in the nucleus accumbens tracks aversive prediction error as it does in reward. To do this, the authors used fast-scan cyclic voltammetry alongside a number of key behavioural manipulations that have been shown to provide conditions for detecting reward prediction error. The behavioral designs used are established and appropriate to the questions posed. While the aversive cue, a loud white noise, employed was not standard, the paper provides behavioural evidence for its effectiveness as an aversive outcome. Although a closer examination of the learning that underlies the task design is necessary. While the analyses of the neural signal are appropriate, some alternatives could be explored in order to better understand the profile of the dopamine signal during the behavioural tasks employed, including evaluation of baseline fluctuations. Relating dopamine concentration to aversive events to that seen to rewarding events will provide important insight into the role of dopamine in general and valence-specific learning mechanisms.

A thorough examination of accumbal dopamine concentration levels to a aversive event and its predictor using some of the key conditions under which a (reward) prediction error is reported. As a result of this thorough investigation the paper is a real tour de force. Some comments/concerns are outlined below.

Interpretation of the data.

The temporal window of DA. Across the experiments DA concentrations are examined during the predictive cue and the WN US and those concentrations are not in line with an error signal. This may be explained in terms of floor effects in the signal, but some aspects of the study argue against that – the signal during the CS for example. The authors have examined post WN period which shows a differential slope to an expected and an unexpected WN, which is in line with a PE account. A more complete analyses of the slope would be worth seeing. A justification for the slope is also necessary. Also, why not take the concentration during a specified post-WN period? The authors could analyze different temporal windows. Importantly, the post-WN period does suggest PE-like differences – most striking in Figure 2A, but potentially present in 2D, E,F.

The baseline subtraction. The authors have used a reasonable baseline, pre-CS period, of determining the change in DA release. However, it would be important to know if and how the baseline changes across the training sessions. It is unclear whether pre-cue baseline subtraction was done on a trial by trial basis or if one averaged baseline was calculated and then used to determine DA concentration change. This could influence the data.

Framing. While I really appreciated the framing of the approach within what has been reported for RPE, I wonder if failing to get the same profile is really conclusive regarding the absence of a PE signal using an aversive event. For example, a complete shift is not reported in all datasets that show a DA RPE signal. This can also be due to variability across animals. Further, it is possible that an aversive event may be tracked more categorically by DA , making the intensity and probability examinations less relevant for testing PE in the aversive case. Further, reduction in DA concentration may also be less sensitive to detecting these subtle changes. Is there anything in the data that can deal away with these points?

Methodology. It seems that a lot (all?) of the rats were ran across all experimental conditions. This raises concern over carry over effects.

Validation of the WN as aversive: The behavioural tasks do not include another auditory cue as a control comparison in the open field nor in the operant. The operant has different intensities of the WN, which helps matters as there is a difference between the 70dB and the 96dB. But it is unclear what the role of the WN is in these conditions. Is it just to show that it is aversive or that it can condition behaviour? I think there is evidence for the former but not the latter.

The manuscript should refer to the WN as an aversive stimulus, not punisher.

Please change all instances that refer to WN intensity (e.g. 70dB, 90dB, etc) as a volume. 'volume' is a colloquial way of referring to intensity and is therefore not appropriate in a scientific setting.

*Reviewer #2 (Recommendations for the authors):*

The manuscript by Goedhoop focuses on understanding the dopaminergic signals that are driven by aversive stimuli. The project uses fast scan cyclic voltammetry to directly record dopamine fluctuations in awake and behaving animals in response to a variety of task variables to parse their contribution to behavior across conditions.

There are a lot of strengths of this manuscript.

First – the use of white noise is innovative and powerful. The field often focuses on aversive footshocks, which are interesting but unique stimuli. The use of white noise allows for an aversive stimulus that is not painful and is not an electrical stimulus which is a significant advantage over previous studies.

Second – one issue in the field in general is that people focus on dopamine as an RPE encoder where dopamine in every context has been linked to RPE-like signaling. However, a shortcoming of previous work with direct dopaminergic recording approaches like voltammetry is that they are electrical in nature and thus, cannot record the response to the aversive stimulus (footshock) themselves. This is clearly and issue as the stimulus response is a critical variable to understand in order to make conclusions about whether something encodes "RPE" or not.

However, even with these strengths there are some significant weaknesses. These occur In both the conceptual presentation and the experimental execution and if addressed the manuscript would be much stronger.

1. Regarding the conceptual issues, the largest is the terminology used throughout the manuscript. One of the major issues in this manuscript is the definition of all aversive stimuli as punishers. A punishment has a specific definition that is incorrectly used here. A punisher is not defined by the valence of the stimulus, but rather the behavioral effect of that stimulus on future behavior. A punisher reduces rates of behavior – appetitive stimuli can also function as punishers. This is a huge problem and the wording in the manuscript should be changed to reflect this. This is incredibly problematic as it suggests that the findings are different than what they actually are on a conceptual level as they relate to what dopamine is doing.

2. There are many statements that are inherently problematic because of this mischaracterization of the behavior. For example: "this heterogeneity is reflected in dopamine responses to punishment throughout the striatum" is stated in the introduction; however, many of these studies are not punishment. Also, many people have suggested that dopamine controls motivational responses. In that case a "punisher" and "negative reinforcer" would show different dopaminergic signatures even though the maintaining stimulus is aversive in both cases. This is actually an important and overlooked aspect of this work and defining everything as a punisher makes it difficult to decipher what the data are showing and how that relates to the actual behavior of the animal.

3. These results can alternatively be explained by the novelty induced alterations of behavior in rodents. The literature has shown that rodents withhold consummatory behavior and novelty induces hyperactivity in rodents (e.g., Bardo et al.,1990; Psychopharmacol; but also see earlier paper from 1950s Berlyne 1955; Bindra and Spinner 1958; Welker 1959). The dopamine system is highly involved in both of these effects.

Regarding experimental issues:

1. The canonical unconditioned aversive response in rodents is freezing or immobility (e.g., Antoniadis and McDonald, 2001, Exp Brain Res). Here in Figure 1 and also in Figure 2C they show the whitenoise itself results in increase in locomotor activity. How do we know this is an aversive response comparable to other traditional aversive stimuli such as footshocks or tail pinches (which are shown to result in increase in NAc core dopamine release see Budygin et al., 2012; Mikhailova et al., 2019).

2. In Figure2A, the dopamine response to the white noise seems to be decreased. However, this is due to the baseline used to compute the white noise dopamine responses, which seems to shift lower due to the dopamine response to the antecedent cue. That is why the initial white noise dopamine response seems to be positive in Trial1 where the baseline is still above 0 but looks negative when the cue response becomes negative starting from Trial3. If the dopamine response to the white noise outcome were computed with a baseline of its own (1-2 sec before the WN outcome) that would result in a positive peak even in trials 11-30.

3. The authors claim that the decrease in dopamine response to the white noise during the first trial of aversive conditioning (Figure 2A) is an unconditioned response. However, there is an immediate positive peak after the white noise presentation on that trial, which lasts about 1 sec. How does the behavior map on to this timeline? Do rats move for the first second but then freeze for the remainder of the white noise presentation? At the very least a strong justification should be made for what is being normalized and if and how you can separate specific task components.

5. There are numerous studies where cues are paired with white noise as an aversive stimulus. It is important to determine if the predictive cue elicited a conditioned response. Without that how do you know the animals made the association? This is important to make conclusions about what the neural signal in response to these cues actually mean.

6. In response to the data with the white noise and different timing. Is this predicting the timing? Or the value of the outcome? These are not dissociable in this experiment and when you discuss timing this would be important to dissociate. This is a critical thing to parse as duration is in the title.

Overall this is an interesting manuscript however in order for it to be suitable for publication the authors should rephrase their terminology to accurately state what the stimuli are and how they relate to behavior as well as make sure to show that white noise does function as an aversive stimulus.

*Reviewer #3 (Recommendations for the authors):*

In this study a talented group of neurochemists performed real-time measurements of dopamine concentration in behaving rats to investigate whether transient accumbal release events encode the value of aversive stimuli. Directly measuring dopamine release events rather than phasic bursts of putative dopamine neural activity is particularly important to determine how transient dopamine signals encode aversive events because recent evidence shows that terminal-terminal modulation influences behaviorally relevant patterns of release that do not necessarily coincide with changes in neural activity. The authors also incorporated an impressive systematic behavioral design and a unique aversive stimulus (i.e., white noise) to address this unresolved controversy. First, they determined that high decibel white noise produced a conditioned place aversion and punished food seeking. Then, by presenting comparable levels of white noise within a Pavlovian context, they found that dopamine release events were suppressed during the presentation an aversive stimulus and its conditioned predictor. They further report that the magnitude by which dopamine release events were suppressed did not correlate with the amplitude of white noise; thereby leading them to conclude that transient dopamine signals in the core region of the nucleus accumbens respond to aversive stimuli, but do not necessarily encode the value of punishment. However, there remain several unresolved issues and points of contention regarding the interpretation of the authors' results. Of note, they did not measure dopamine release during punished behavior, but rather in the presence of an aversive stimulus that increased the behavior being assessed. In addition, it is not clear whether the rats were able to discriminate between the tightly dispersed decibels of white noise presented during the Pavlovian task in which dopamine concentration was measured. While the current results are intriguing and a technical advance over preceding electrochemical studies, the overall picture of how transient dopamine signals throughout the mesocorticolimbic pathway encode aversive stimuli still requires further clarification that the current group of authors are capable of providing.

The submission includes an excellent set of well-considered experiments; I am both impressed and intrigued. However, I do have some constructive criticism, suggestions, and alternative interpretations to consider.

A timeline or illustration of the different subgroups and conditions under which FSCV recordings occurred would increase the readability of the manuscript.

If the authors do not believe they are measuring dopamine (DA) value signals associated with aversive stimuli, have they considered whether they are measuring a correlate of the acoustic startle response? Acoustic startle is commonly associated with an increase in ambulation (as reported in the current manuscript). This alternative interpretation would provide an important missing piece of data from previously hypothesized neural circuitry underlying acoustic startle (see figure 6 of Koch and Schnitzler, 1997). Furthermore, the transient accumbal DA signal has previously been associated with pre-attentive sensory perception of salience. Might your results align more with Redgrave's work demonstrating that there are indeed distinct DA sensory responses that are dissociable from accumbal value signals; possibly also involved in the acoustic startle response?

Koch M, Schnitzler HU. The acoustic startle response in rats-circuits mediating evocation, inhibition and potentiation. Behavioural brain research. 1997 Dec 1;89(1-2):35-49.

The authors should discuss the current results in the context their previous work (specifically DA correlates with 22Kh USVs) with the Wohr lab, which was surprisingly not referenced in the current manuscript.

Willuhn I, Tose A, Wanat MJ, Hart AS, Hollon NG, Phillips PE, Schwarting RK, Wöhr M. Phasic dopamine release in the nucleus accumbens in response to pro-social 50 kHz ultrasonic vocalizations in rats. Journal of Neuroscience. 2014 Aug 6;34(32):10616-23.

It is not clear whether rats could actually discriminate between the different tightly dispersed white noise volumes in the Pavlovian task during FSCV measurements-a centrally important experiment to support the authors' conclusions about dopamine and value. Based on my current interpretation of the methods, rats were able to discriminate between the white noise volumes in the operant choice task, but the conditions were substantially different from those in which the FSCV recordings occurred. Aside from being instrumental rather than Pavlovian, five consecutive training sessions occurred for each of three white noise volumes (70, 90, 96db) before discrimination testing was tested; and approximately 100 trials occurred in each session, with a single white noise volume being tested per session. Then, during volume-response FSCV recordings, each of four white noise volumes (70, 80, 90, 96db) were randomly played 12 times each in a single session. In a separate group of recordings, the authors performed FSCV recordings in the presence of 80 vs. 96db white noise volumes. Compared to the volume-response FSCV recordings, more training and trials occurred but, the Pavlovian trials were still randomly presented within a session. Thus, I caution against assuming that the animals could discriminate between the less dispersed and randomly presented white noise volumes presented in the single session Pavlovian experiments (particularly those depicted in 2E) based on the results from a methodologically distinct instrumental choice experiment. The authors should further address (either experimentally or logically) why the reader should accept that the rats could discriminate between the tightly spaced decibel volumes (particularly in the volume-response experiment).

Please provide additional clarification on figure 2B and your interpretations of it. First, the dopamine response to the CS predicting the aversive stimulus does seem to increase with experience (albeit quickly), which I contend contradicts your statement starting on line 293: 'Although the WN-predictive, conditioned cue acquired the ability to reliably suppress NAC dopamine release, no substantial transfer of this effect from US to CS occurred a prerequisite for a prediction error signal.' Aversive stimuli are known to rapidly induce conditioned responses. For example, a conditioned fear response is often established in a single fear conditioning session with just a few pairings of the CS and aversive stimuli. Thus, I contend (as the authors generally indicate in the results) that the US to CS transfer does indeed occur during aversive conditioning, it just occurred rapidly (on day 1; as would occur in standard fear conditioning). Similar to the presentation of extinction data in 2A, what do the CS and US data look like trial-by-trial on day 1 of aversive conditioning?

Please provide an explanation regarding the degrees of freedom associated with the statistics used for your comparisons in figure 2B (lines 242-244). It is unclear to me how the data points from the top figures transfer to the bottom figures and the degrees of freedom are adding to my confusion.

Also, what affect did smoothing the data with a 10-point median filter prior to analysis have on the results? While smoothing the data for visual presentation is common, I question whether performing statistical analysis on the data after smoothing it might affect the results?

Furthermore, the maximal amplitude of the CS associated DA response during appetitive conditioning appears to be comparable between day 1 and day 6 in the top right panel of 2B. Thus, it seems that the duration of the signal, rather than it's amplitude, is responsible for the significant effect shown in the bar graph of the bottom panel of 2B. From this observation, is the duration of the signal not accounting for the majority of appetitive value coding? Did smoothing the data contribute to the longer duration?

The authors found that a longer duration of white-noise exposure produced a longer suppression in dopamine release. One could argue that both the frequency and amplitude of a signal should influence neural coding. Thus, why would a prolonged reduction in frequency not be reflective of greater aversive value? Why do the authors exclusively consider the amplitude of their dopamine in the context of aversive value determinations?

The use of white noise as an aversive stimulus is championed by the authors because it does not produce the behavioral confound of freezing observed in standard fear conditioning approaches using electrical footshock. However, white noise and its conditioned predictor increased ambulatory behavior in the current study. Thus, how is the logic of avoiding behavioral DA responses not flawed? Could DA responses correlate to the initiation of action (which could be dissociable from speed) not confound DA value coding assessments across different volumes of white noise?

I also have some related critiques to consider regarding the benefits of white noise espoused in the discussion, starting on line 435. I already pointed out that the logic of using white noise because it doesn't induce freezing and thereby avoids movement-DA related confounds is flawed as white noise increased ambulation-which again could confound DA value coding of aversive stimuli if indeed DA transients are directly related to movement. But the data from the current study and others (PMID: 24345819, figure 2) might suggest that accumbal transients are not actually correlated in a positive way to general increases in activity. Regardless, I also take issue with point 2 and 3 (line 438) because foot shock can also maintain avoidance across many trials and sessions and be titrated with varying valence and contingencies; what are you contrasting white noise too? Point 4 does not remove the potential confound of Redgrave's work, as he has repeatedly demonstrated that the accumbal transient DA response can be induced by 'pre-attentive' subcortical sensory input. Point 6 is also not exactly accurate as properly isolating the electrical components of a shock generator eliminates the noise artifacts it can produce; thus, white noise might be easier but I don't think it is fair to imply that properly set-up electrical foot shock interferes with or jeopardizes FSCV recordings. It is fair that electrical artifacts are detected during single-unit recordings but they are easy to detect and remove during sorting.

The study would be strengthened by a core vs. shell vs. PFC comparison but at the very least, the literature regarding the role of dopamine in these distinct regions should be addressed in the discussion. I acknowledge that performing measurements of DA release in the PFC is a fraught endeavor but, at this level of journal I would expect that you at least address the Tye lab's data on aversive stimuli and the PFC (generally reviewed in: Vander Weele et al. 2019). Is it possible that DA signals in the PFC but not the NAc core encode aversive value? You also point out that Badrinarayan et al., 2012 reported that the same CS that reduced dopamine in the core increased DA in the shell-a somewhat paradoxical finding that was neither explored in the current study nor addressed in the discussion. While it is an important replication to show that NAc DA in the core is reduced by conditioned predictors of aversive stimuli, and the analysis done in figure 3 is an impressive advance over previous studies, I am left questioning the advance provided by the current data set. The primary positive effect in the core is a replication. Building a story on the general role transient dopamine signals play in encoding aversive stimuli using negative FSCV effects that were determined using tightly spaced decibels of white noise that the rat may not have been able to discriminate is shaky.

Vander Weele CM, Siciliano CA, Tye KM. Dopamine tunes prefrontal outputs to orchestrate aversive processing. Brain research. 2019 Jun 15;1713:16-31

I strongly suggest that the authors not use the term punishment in the context of their results and instead use a term such as aversive stimulus. While multiple definitions of punishment exist in the literature, the one that is almost universally taught in the psychological context of animal behavior today is that of Azrin and Holz (1966), which considers punishment as a reduction in behavior in response to a stimulus. According to this definition, the behavioral response is the key element in determining whether you are observing punishment or a reinforcement; with punishment describing a decrease in behavior and reinforcement describing an increase in behavior. It is additionally worth considering that an aversive stimulus (e.g., electrical shock) can function as either a punisher or a reinforcer (i.e., something that increases behavior) in the operant context (McKearny, 1966; Morse and McKearney 1977), during imprinting (Hess, 1959), or in human interaction (Mello 1978; Sack and Miller, 1975). In the current study white noise does seem to punish food-maintained responding in one context, but also increases ambulatory behavior in another context.

Azrin NH, Holz WC. Punishment. Operant behavior: Areas of research and application. 1966:380-447

McKearney JW. Maintenance of responding under a fixed-interval schedule of electric shock-presentation. Science. 1968 Jun 14;160(3833):1249-51.

Morse WH, McKearney JW, Kelleher RT. Control of behavior by noxious stimuli. In Handbook of psychopharmacology 1977 (pp. 151-180). Springer, Boston, MA.

Hess EH. Imprinting. Science. 1959 Jul 17;130(3368):133-41.

Mello NK. Control of drug self-administration: The role of aversive consequences. Phencyclidine (PCP) Abuse: An appraisal. National Institute on Drug Abuse Research Monograph. 1978 Aug 1(21):289-308.

Sack RL, Miller W. Masochism: A clinical and theoretical overview. Psychiatry. 1975 Aug 1;38(3):244-57.

Along this same line of thought, could it not be concluded that DA might scale with the value of punishment but not of aversive stimuli that do not actually reduce the occurrence of a behavior? Punishment-associated DA signals in the basal ganglia might be more correlated to stimuli that actually reduce behavior given this neural circuit's well-determined role in goal-directed learning. Thus, if you want to conclude that DA fails to encode punishment (rather than an aversive stimulus), I would want to actually see transient DA signals failing to correlate to stimuli that reduce behavior to different magnitudes.

[Editors’ note: further revisions were suggested prior to acceptance, as described below.]

Thank you for resubmitting the paper entitled "Nucleus-accumbens dopamine tracks aversive stimulus duration and prediction but not value or prediction error" for further consideration by *eLife*. Your revised article has been evaluated by a Senior Editor and a Reviewing Editor. We are sorry to say that we have decided that this submission will not be considered further for publication by *eLife*.

During the review process, the editors and reviewers evaluated and discussed your revised manuscript and your responses to the initial concerns. Unfortunately, all agreed that the data provided are insufficient to convince that the white noise is aversive. The data in Figure 1A and B that are used to make this point do not have the necessary controls. Further, db do not scale linearly. Therefore, the level of aversiveness of the different db of white noise also needs to be verified behaviourally. The valence of the white noise is a the backbone to the story of the paper and the absence of strong evidence that speaks to this issue within the paper was judged to be problematic, precluding it from further consideration for publication. We're sure this is not the decision you were hoping for, but appreciate the chance to reconsider your manuscript and hope that our evaluation will be useful for you as you move forward with this work.

[Editors’ note: further revisions were suggested prior to acceptance, as described below.]

Thank you for resubmitting your work entitled "Nucleus-accumbens dopamine tracks aversive-stimulus duration and prediction but not value or prediction error" for further consideration by *eLife*. Your revised article has been evaluated by Kate Wassum (Senior Editor) and a Reviewing Editor.

The manuscript has been improved but there are some remaining issues that need to be addressed, as outlined below:

*Reviewer #1 (Recommendations for the authors):*

The authors did an excellent job addressing my previous concerns. Of note, their additional data provide convincing evidence that animals could discriminate the different db of white noise and that all intensities function as aversive stimuli in naive rats. The additional locomotor control experiments also strengthen the manuscript. The text is also significantly improved, particularly regarding the concept of punishment. I found the experiments to be more intriguing with better framing. The new additional summary table helped me navigate the manuscript and, importantly, illustrates the impressive breadth of experiments conducted to address a contentious and important question in the field.

*Reviewer #2 (Recommendations for the authors):*

The authors have done a good job at responding to previous comments. I do think that the study is interesting and important for the field. I have a few additional comments that should be addressed.

Several manuscripts have come out recently specifically looking at dopamine release and aversive associative learning. These are surprisingly not cited or mentioned at all in the current manuscript and are highly relevant to the current work (Kutlu et al., 2022, Nature Neuroscience; Kutlu et al., 2021,. Current Biology). Both of these studies record dopamine release in the NAc core during aversive conditioning and relate dopamine signals to aversive stimulus responses and omissions based on previous predictions.

It would be interesting and important for the authors to discuss how aversive stimuli that induce different unconditioned responses – freezing, vs increased motor activity – could relate to dopamine signatures that they induce. Would the authors expect that dopamine responses to aversive stimuli that induce freezing be opposite to those that drive increases in activity?

---

## [Author Response]

[Editors’ note: the authors resubmitted a revised version of the paper for consideration. What follows is the authors’ response to the first round of review.]

Reviewer #1 (Recommendations for the authors):Understanding the role of dopamine beyond reward and specifically in aversion is of fundamental importance. The paper set out to examine whether dopamine concentration in the nucleus accumbens tracks aversive prediction error as it does in reward. To do this, the authors used fast-scan cyclic voltammetry alongside a number of key behavioural manipulations that have been shown to provide conditions for detecting reward prediction error. The behavioral designs used are established and appropriate to the questions posed. While the aversive cue, a loud white noise, employed was not standard, the paper provides behavioural evidence for its effectiveness as an aversive outcome. Although a closer examination of the learning that underlies the task design is necessary. While the analyses of the neural signal are appropriate, some alternatives could be explored in order to better understand the profile of the dopamine signal during the behavioural tasks employed, including evaluation of baseline fluctuations. Relating dopamine concentration to aversive events to that seen to rewarding events will provide important insight into the role of dopamine in general and valence-specific learning mechanisms.A thorough examination of accumbal dopamine concentration levels to a aversive event and its predictor using some of the key conditions under which a (reward) prediction error is reported. As a result of this thorough investigation the paper is a real tour de force. Some comments/concerns are outlined below.Interpretation of the data.The temporal window of DA. Across the experiments DA concentrations are examined during the predictive cue and the WN US and those concentrations are not in line with an error signal. This may be explained in terms of floor effects in the signal, but some aspects of the study argue against that – the signal during the CS for example. The authors have examined post WN period which shows a differential slope to an expected and an unexpected WN, which is in line with a PE account. A more complete analyses of the slope would be worth seeing. A justification for the slope is also necessary. Also, why not take the concentration during a specified post-WN period? The authors could analyze different temporal windows. Importantly, the post-WN period does suggest PE-like differences – most striking in Figure 2A, but potentially present in 2D, E,F.This (no APE signal) may be explained in terms of floor effects in the signal, but some aspects of the study argue against that – the signal during the CS for example.

The reviewer expresses a very reasonable concern, but also points out that we present data that speaks against the floor-effect hypothesis: (1) In Figure 3C, we show that the overall dopamine decrease during CS+US is greater than during CS alone. This would not be the case if a floor effect was present during the 6 s WN presentation. (2) Figures3D and 3G demonstrate that when WN duration is extended, dopamine concentration continues to decrease further than what is seen for the 6 s WN we applied in most of the experiments. Thus, at no point did we detect a floor effect. Moreover, even in the presence of a hypothetical floor effect, the WN-dopamine response would still not align with an aversive-prediction error (APE), because in Figure 3E, the dopamine recovery slope in case of a better than-expected scenario (CS alone) is the same as for the expected scenario (CS+WN) (e.g., no observable rebound that encodes an “error”).

A justification for the slope is also necessary.

Our justification for using the slope in Figures3D-F as a readout for dopamine dynamics, is that it is more sensitive and informative for probing dopamine function in the framework of an APE (based on the results reported in figures prior) than averaging dopamine concentration over a defined time window/epoch (which we otherwise employed), or alternatives employed in other studies, such as area under the curve or peak concentration. Using the slope allows for integration of the change in dopamine concentration over time, as opposed to averaging concentrations over an epoch (in which there is no integration over time). Averaging over an epoch would be insensitive to whether dopamine rebound recovers rapidly or slowly, and unable to determine when (during the averaged epoch) a dopamine change takes place. This type of information is crucial in Figures3D-F, as it allows us to differentiate between tracking an aversive stimulus and encoding of an APE: Averaging the changes in dopamine concentration would incorrectly lead to the interpretation that dopamine encodes prediction errors, even though the data can be explained by a much easier computation, i.e. the tracking of WN or its predictor (the data in Figure 3F is an indirect exception, as dopamine concentration recovers slower following unpredicted WN). Thus, in this case, averaging over an epoch would yield false positive results and would have failed to detect the tracking. This level of information depth is not required to draw conclusions in scenarios where slope does not differ, as for example in Figure 2A. In conclusion, assessing slopes in dopamine-concentration change provides another level of information.

Also, why not take the concentration during a specified post-WN period? The authors could analyze different temporal windows.

The reviewer is correct, we could use different temporal windows to achieve similar sensitivity as for the slope analysis. However, there is no advantage in doing so compared to the slope. In fact, it would be less objective and more complicated (e.g., determining the size of appropriate temporal windows), as well as being less powerful statistically. Thus, we believe that the slope analysis provides a superior readout.

“Importantly, the post-WN period does suggest PE-like differences – most striking in Figure 2A, but potentially present in 2D, E,F.

It is unclear to us how Figure 2A suggests evidence for PE-like differences. It simply shows that the concentration goes down during WN onset and (later in the session) also in response to the CS predicting WN. And after WN cessation the concentration goes up again (post-WN period). Although these data are not completely inconsistent with an APE per se, they are better explained by the “tracking explanation”, as the dopamine dips do not shift over to the CS substantially (as analyzed in Figure 2B left), thus speaking against an APE (even before considering the additional evidence against an APE provided in later figures). Furthermore, statistical analysis demonstrates that there are no significant slope differences in Figure 2A.

Similarly, Figures2D-F (now Figures2D,E,G in the new version of the manuscript) do not provide convincing evidence of an APE. Particularly, experiments describing different WN intensities in Figures2E and G (formerly F) speak against an APE, since a prediction error should by definition respond to different outcome values.

We think, that the reviewer may be referring to “aversive prediction” (a function restricted to reacting to cues that predict an aversive stimulus, without reacting to prediction errors) instead of APE, since there is certainly strong evidence for aversive prediction (i.e., the CS acquires the ability to decrease dopamine on its own) in the figures that the reviewer refers to.

The baseline subtraction. The authors have used a reasonable baseline, pre-CS period, of determining the change in DA release. However, it would be important to know if and how the baseline changes across the training sessions. It is unclear whether pre-cue baseline subtraction was done on a trial by trial basis or if one averaged baseline was calculated and then used to determine DA concentration change. This could influence the data.

The reviewer addresses an important point. Pre-cue baseline subtraction was performed on a trial-by-trial basis, which minimizes the effects of changes over time (including across training sessions). Therefore, baseline changes across training sessions did not significantly influence the reported outcomes. This is further underlined by the fact that pre-cue baseline was stable for the most part, and did not drift up or down significantly over time (see the horizontal pre-event dopamine traces in all figures). We have added information to the Methods section to clarify how we performed the baseline subtraction.

While we completely agree that it would be very interesting to record baseline changes across training sessions, this lies both beyond the technical possibilities of the experiments conducted, as well as beyond the scope of our study. APE- or tracking-like changes occur on a short time scale (seconds). The only baseline difference that could hypothetically influence the experimental outcomes would be the above-mentioned floor effect, which, as we pointed out above, did not occur.

Framing. While I really appreciated the framing of the approach within what has been reported for RPE, I wonder if failing to get the same profile is really conclusive regarding the absence of a PE signal using an aversive event. For example, a complete shift is not reported in all datasets that show a DA RPE signal. This can also be due to variability across animals. Further, it is possible that an aversive event may be tracked more categorically by DA , making the intensity and probability examinations less relevant for testing PE in the aversive case. Further, reduction in DA concentration may also be less sensitive to detecting these subtle changes. Is there anything in the data that can deal away with these points?I wonder if failing to get the same profile is really conclusive regarding the absence of a PE signal using an aversive event.

The reviewer brings up a good methodological point. An axiomatic approach as repeatedly validated and performed by Rutledge, Glimcher and colleagues is arguably the most objective way to formally identify a prediction error (Caplin et al., 2010; Hart et al., 2014), because it tests several fundamental assumptions regarding prediction errors systematically and simultaneously. We applied this approach to aversive-stimulus prediction as reported in Figures2G and H and Figures3A-C. Neither the results of this approach nor any other of our results support the hypothesis that an APE is encoded by NAC dopamine. We therefore believe it justified to conclude that WN-induced changes are best described by the “tracking explanation” than an “APE explanation” (especially since “tracking” describes dopamine dynamics well).

Caplin A, Dean M, Glimcher PW, Rutledge RB. Measuring beliefs and rewards: A neuroeconomic approach. Q J Econ. 2010 Dec 31;125(3):923960.

Hart AS, Rutledge RB, Glimcher PW, Phillips PE. Phasic dopamine release in the rat nucleus accumbens symmetrically encodes a reward prediction error term. J Neurosci. 2014 Jan 15;34(3):698-704.

For example, a complete shift is not reported in all datasets that show a DA RPE signal. This can also be due to variability across animals.

Although the reviewer is correct that perhaps not all reported datasets exhibit a complete dopamine RPE shift, we are not aware of any dataset (recorded with a similar technique) demonstrating as little of a shift as reported in our manuscript for WN-cue conditioning (e.g., Day et al., 2007; Clark et al., 2010; Brown et al., 2011; Hollon et al., 2014; not even in so-called goal-tracking animals (our experimental setup did not allow for goal-tracking): Flagel et al., 2011). Moreover, as a control, we provide reward-conditioning related dopamine measurements that were taken under almost identical circumstances as our WN data, that demonstrate a much more substantial shift to the cue (Figure 2B, right). The data from these animals displays very little variance. Importantly, a complete shift is not the point we intended to make (since this can be dependent on training duration); rather the point is that RPEs shift substantially more than the reported WN data with comparable training amount. In addition, to extend on the topic of training duration: For some of the experiments, we trained animals for 9 days (Figure 3B and C), which is a very substantial amount of training.

Day JJ, Roitman MF, Wightman RM, Carelli RM. Associative learning mediates dynamic shifts in dopamine signaling in the nucleus accumbens. Nat Neurosci. 2007 Aug;10(8):1020-8.

Clark JJ, Sandberg SG, Wanat MJ, Gan JO, Horne EA, Hart AS, Akers CA, Parker JG, Willuhn I, Martinez V, Evans SB, Stella N, Phillips PE. Chronic microsensors for longitudinal, subsecond dopamine detection in behaving animals. Nat Methods. 2010 Feb;7(2):126-9.

Brown HD, McCutcheon JE, Cone JJ, Ragozzino ME, Roitman MF. Primary food reward and reward-predictive stimuli evoke different patterns of phasic dopamine signaling throughout the striatum. Eur J Neurosci. 2011; 34(12):1997-2006.

Hollon, N. G., Arnold, M. M., Gan, J. O., Walton, M. E. and Phillips, P. E. M. Dopamine-associated cached values are not sufficient as the basis for action selection. Proc. Natl. Acad. Sci. U. S. A. 111, 18357–18362 (2014).

Flagel SB, Clark JJ, Robinson TE, Mayo L, Czuj A, Willuhn I, Akers CA, Clinton SM, Phillips PE, Akil H. A selective role for dopamine in stimulusreward learning. Nature. 2011 Jan 6;469(7328):53-7.

…aversive event may be tracked more categorically by DA, making the intensity and probability examinations less relevant for testing PE in the aversive case.

We believe that the reviewer is making the exact point we are trying to convey: Aversive events are tracked more categorically by dopamine. It “only” tracks the presence of an aversive stimulus or its predictive stimulus via a slow but constant decrease in NAC dopamine concentration, irrespective of the size of the prediction error experienced. This is one example for how our findings are inconsistent with a formal prediction error, which is generally conceptualized as a quantitative discrepancy between the outcome expected and the outcome experienced and, thus, scales with the relative value of the encoded stimulus and unexpected deviations from this value (BrombergMartin and Hikosaka, 2009; Nasser et al., 2018). We believe this is sufficient evidence to rule out encoding of that what is commonly defined as a prediction error, and that this is a noteworthy finding, as very few studies have addressed this issue systematically.

Bromberg-Martin, E.S., and Hikosaka, O. (2009). Midbrain dopamine neurons signal preference for advance information about upcoming rewards. Neuron, 63(1), 119-126.

Nasser HM, Lafferty DS, Lesser EN, Bacharach SZ, Calu DJ (2018). Disconnection of basolateral amygdala and insular cortex disrupts conditioned approach in Pavlovian lever autoshaping. Neurobiol Learn Mem. 147:35-45.

Reduction in DA concentration may also be less sensitive to detecting these subtle changes.

Unfortunately, the meaning of this sentence is unclear to us. Is the reviewer referring to reductions as opposed to increases in dopamine concentration? Or that the measurement of reductions is more difficult technically? With regards to the latter, the applied technique, fast-scan cyclic voltammetry (FSCV), is very capable of detecting both (phasic) increases and decreases in extracellular dopamine concentration, as evidenced by Figure 1G. With regards to the former, we believe that we may have addressed the concern already in our discussion: “The dopamine system presumably has a bigger dynamic range for increasing activity; it can do so, for example, by increasing the number of cells firing and their firing frequency (and thereby the total number of dopamine-containing vesicles being released). In contrast, dopamine-signaling reduction cannot drop below a certain point, since the cells’ maximum response is to cease firing altogether, and extracellular dopamine can only be removed relatively slowly or must diffuse away. This disparity could translate into a structurally-limiting factor on what can be encoded by a reduction in dopamine concentration and explain some of the above-mentioned differences in function. However, the slow-ramping declining and recovery slopes we observed do not reflect the system limits, since the very first exposure to WN resulted in a steeper decline and rewards given during WN resulted in steeper increases.”

Methodology. It seems that a lot (all?) of the rats were ran across all experimental conditions. This raises concern over carry over effects.

Although we understand concerns relating to carry-over effects in animal behavioral paradigms in general, we are unsure what kind of carry-over effects the reviewer is referring to in the context of our experiments that may compromise our data. Even if there was such an effect conceivable, it would not affect our results, as all comparisons are made within-session. Besides, the initial multi-day cue conditioning reveals dopamine signals maintain the same size between days 1 and 6, thus, no habituation takes place in this context. Furthermore, we now provide new, additional data demonstrating that animals differentiate between different WN intensities, even after many sessions of WN exposure (Figure 2F). Finally, Figures 3D-F demonstrate that dopamine traces induced by WN still look similar to day 1 of exposure, yet change readily when experimental conditions are altered.

On the contrary, we believe our approach of re-using animals between experiments in fact increases consistency between experiments (especially regarding concerns about interindividual differences). Besides this, there is the ethical advantage of reducing the number of animals used.

However, we may address the reviewer’s concern about carry-over effects quantitatively by calculating data stability across time:

**Author response image 1. sa2fig1:** Cue and WN induce dopamine decrease across repeated experiments in the same animals. Average dopamine concentration during cue and WN epochs in the animals depicted in Figures2A (first 30 trials on day 1 of conditioning), 2G (deterministic experiment), and 3B and C (probabilistic experiment). There is a significant main effect (F(1.428, 7.141) = 9.402, p = 0.0133) of recording session during the cue epoch, and post-hoc testing determined there is a significant difference between day 1 of conditioning and the deterministic experiment (p = 0.0035; Tukey’s multiple comparisons test). However, we found no effect of recording session on dopamine concentration during the WN epoch (F(1.236, 6.178) = 0.03898, p = 0.8936) indicating that the neurochemical response to WN remains stable across days.

Validation of the WN as aversive: The behavioural tasks do not include another auditory cue as a control comparison in the open field nor in the operant. The operant has different intensities of the WN, which helps matters as there is a difference between the 70dB and the 96dB. But it is unclear what the role of the WN is in these conditions. Is it just to show that it is aversive or that it can condition behaviour? I think there is evidence for the former but not the latter.The behavioural tasks do not include another auditory cue as a control comparison in the open field nor in the operant [box]”

The reviewer makes a good suggestion: Indeed, adding a control stimulus would enable useful comparisons. However, we do not believe that the lack of these control comparisons undermines our conclusions, since the animals avoid WN at higher intensities more than at lower intensities (Figure 1C; operant box), and in new data we show that animals differentiate between different WN intensities as exemplified by increasing locomotion with higher WN intensities (Figure 2F; operant box). A cue predictive of WN only slowly acquires locomotion-inducing effects with multiple pairings (Figure 2C, left). Once this cue is no longer predictive of WN, it loses its dopamine-reducing effects (Figure 2D). Finally, the auditory cues used here are standard sounds produced by commercially available and widely used Med Associates operant boxes, which have been used in a plethora of studies without obvious aversive effects.

Above all, we did not investigate what kind of sound or which aspect of WN is aversive. We investigated how the dopamine system reacts to an aversive stimulus and its prediction.

Is it just to show that it is aversive or that it can condition behaviour?

It is to show both. Our interpretation is that WN is not just aversive, but its conditioning to a cue can also elicit conditioned behavior. Figure 2C shows a cue-induced increase in speed (green trace during cue presentation from 0-5 s), which is a conditioned response.

The manuscript should refer to the WN as an aversive stimulus, not punisher.

We agree with the reviewer and have changed the manuscript accordingly.

Please change all instances that refer to WN intensity (e.g. 70dB, 90dB, etc) as a volume. 'volume' is a colloquial way of referring to intensity and is therefore not appropriate in a scientific setting.

We changed the manuscript accordingly (“intensity” instead of “volume”).

Reviewer #2 (Recommendations for the authors):The manuscript by Goedhoop focuses on understanding the dopaminergic signals that are driven by aversive stimuli. The project uses fast scan cyclic voltammetry to directly record dopamine fluctuations in awake and behaving animals in response to a variety of task variables to parse their contribution to behavior across conditions.There are a lot of strengths of this manuscript.First – the use of white noise is innovative and powerful. The field often focuses on aversive footshocks, which are interesting but unique stimuli. The use of white noise allows for an aversive stimulus that is not painful and is not an electrical stimulus which is a significant advantage over previous studies.Second – one issue in the field in general is that people focus on dopamine as an RPE encoder where dopamine in every context has been linked to RPE-like signaling. However, a shortcoming of previous work with direct dopaminergic recording approaches like voltammetry is that they are electrical in nature and thus, cannot record the response to the aversive stimulus (footshock) themselves. This is clearly and issue as the stimulus response is a critical variable to understand in order to make conclusions about whether something encodes "RPE" or not.However, even with these strengths there are some significant weaknesses. These occur In both the conceptual presentation and the experimental execution and if addressed the manuscript would be much stronger.1. Regarding the conceptual issues, the largest is the terminology used throughout the manuscript. One of the major issues in this manuscript is the definition of all aversive stimuli as punishers. A punishment has a specific definition that is incorrectly used here. A punisher is not defined by the valence of the stimulus, but rather the behavioral effect of that stimulus on future behavior. A punisher reduces rates of behavior – appetitive stimuli can also function as punishers. This is a huge problem and the wording in the manuscript should be changed to reflect this. This is incredibly problematic as it suggests that the findings are different than what they actually are on a conceptual level as they relate to what dopamine is doing.

The reviewer is correct. Although we were aware that our use of the word “punishment” was not in the pure sense of its definition in Psychology, and although we stated in the *Introduction* of the previous version of the manuscript that: “we will frequently refer to aversive stimuli as punishments, irrespective of whether conditioning was of instrumental or Pavlovian nature.”, we agree it was a suboptimal use of the word “punishment”. We understand the reviewer’s concern about the deviation from a definition based on the reduction rate of behavior. We have changed the manuscript accordingly to reflect that we indeed intend to describe and focus on the valence of the stimulus. We replaced the word “punishment” with “aversive stimulus” and the term “punishment-prediction error” with “aversive prediction error”. However, as much as we agree that this was a suboptimal choice in terminology, we do not share the reviewer’s opinion that this was a “huge problem” and “incredibly problematic”: Regardless of terminology, the data convincingly reflect the dopamine response to aversive white noise, and our observations on its contrast to dopamine dynamics around rewards are not compromised.

2. There are many statements that are inherently problematic because of this mischaracterization of the behavior. For example: "this heterogeneity is reflected in dopamine responses to punishment throughout the striatum" is stated in the introduction; however, many of these studies are not punishment. Also, many people have suggested that dopamine controls motivational responses. In that case a "punisher" and "negative reinforcer" would show different dopaminergic signatures even though the maintaining stimulus is aversive in both cases. This is actually an important and overlooked aspect of this work and defining everything as a punisher makes it difficult to decipher what the data are showing and how that relates to the actual behavior of the animal.

We agree with the reviewer. As stated above under point 1, we replaced the word “punishment” with “aversive stimulus”. We believe that has appropriately removed what the reviewer refers to as “mischaracterization of the behavior” and guides the focus of our manuscript to the valence of the WN stimulus. By refining our definition, we have also addressed the reviewers second point: the data reflect dopamine response and animal behavior associated with aversive stimuli. We agree with the reviewer that terminology is important to communicate our findings appropriately.

3. These results can alternatively be explained by the novelty induced alterations of behavior in rodents. The literature has shown that rodents withhold consummatory behavior and novelty induces hyperactivity in rodents (e.g., Bardo et al.,1990; Psychopharmacol; but also see earlier paper from 1950s Berlyne 1955; Bindra and Spinner 1958; Welker 1959). The dopamine system is highly involved in both of these effects.

The reviewer posits an interesting alternative hypothesis to explain some of our data. However, although we agree that novelty may also induce increased locomotion and although we cannot exclude that novelty may have played a role in the first few presentations of the WN, we believe that novelty does not explain our findings:

1) Novelty is defined as “the quality of being new and unusual” (https://dictionary.apa.org/novelty). The reviewer cites several articles that support the conclusion that novelty vanishes on the scale of minutes (Welker, 1959; Berlyne, 1955; Bindra and Spinner 1958). Our rats, however, were exposed to WN hundreds of times, over the course of up to 10 sessions, with each a duration of 1.5 hours, and with (on average) approximately 66 exposures per session; thus, to these rats, WN cannot be described as novel, new, or unusual. We see this reflected in the dopamine response, which is stable across many sessions with many WN presentations.

(As a side note: Bardo et al. (1990) studied the effect of dopaminergic drugs on locomotion in a novel environment, but did not compare to a non-novel environment).

2) We report that locomotion and dopamine are not correlated (linearly; Figure 2H).

3) Novelty tends to induce locomotion towards a novel object or sensory stimulus, whereas the animals in our study avoid the WN stimulus (Figure 1A).

4) We are unsure why the reviewer brings up the withholding of consummatory behavior as an alternative explanation or confounding factor, since our experiments do not deal with withholding of consummatory behavior. Thus, the reported decrease in dopamine concentration cannot be explained by stopping consummatory behavior. 5) Dopamine release is not tied to consummatory behavior itself, as has been shown by a number of studies that demonstrate a temporal shift in dopamine release away from consummatory behavior, to the cue that predicts foodreward delivery (e.g., Day et al., 2007; Clark et al., 2010; Brown et al., 2011; Hollon et al., 2014).

Berlyne DE. The arousal and satiation of perceptual curiosity in the rat. J Comp Physiol Psychol. 1955 Aug;48(4):238-46.

D Bindra, N Spinner. Response to different degrees of novelty: the incidence of various activities. J Exp Anal Behav. 1958 Oct;1(4):341-50. – Welker WI. Escape, exploratory, and food-seeking responses of rats in a novel situation. J Comp Physiol Psychol. 1959 Feb;52(1):106-11. – MT Bardo 1, SL Bowling, RC Pierce. Changes in locomotion and dopamine neurotransmission following amphetamine, haloperidol, and exposure to novel environmental stimuli. Psychopharmacology (Berl). 1990;101(3):338-43.

Day JJ, Roitman MF, Wightman RM, Carelli RM. Associative learning mediates dynamic shifts in dopamine signaling in the nucleus accumbens. Nat Neurosci. 2007 Aug;10(8):1020-8.

Clark JJ, Sandberg SG, Wanat MJ, Gan JO, Horne EA, Hart AS, Akers CA, Parker JG, Willuhn I, Martinez V, Evans SB, Stella N, Phillips PE. Chronic microsensors for longitudinal, subsecond dopamine detection in behaving animals. Nat Methods. 2010 Feb;7(2):126-9.

Brown HD, McCutcheon JE, Cone JJ, Ragozzino ME, Roitman MF. Primary food reward and reward-predictive stimuli evoke different patterns of phasic dopamine signaling throughout the striatum. Eur J Neurosci. 2011; 34(12):1997-2006.

Hollon, N. G., Arnold, M. M., Gan, J. O., Walton, M. E. and Phillips, P. E. M. Dopamine-associated cached values are not sufficient as the basis for action selection. Proc. Natl. Acad. Sci. U. S. A. 111, 18357–18362 (2014).

Regarding experimental issues:1. The canonical unconditioned aversive response in rodents is freezing or immobility (e.g., Antoniadis and McDonald, 2001, Exp Brain Res). Here in Figure 1 and also in Figure 2C they show the whitenoise itself results in increase in locomotor activity. How do we know this is an aversive response comparable to other traditional aversive stimuli such as footshocks or tail pinches (which are shown to result in increase in NAc core dopamine release see Budygin et al., 2012; Mikhailova et al., 2019).

As stated by Antoniadis and McDonald (2001), somatomotor immobility (freezing) is a common fear response in many species, however, the same paper also states that another common fear response is withdrawal (avoidance or escape) from the danger, which includes increased locomotion. In fact, unconditioned aversive response in rodents depends on the situation/environment and aversiveness of the stimulus. As Schoonover et al. (2017) state: “Rodents exhibit a variety of defensive behaviors in response to innately aversive and conditioned cues, including flight, freezing, crouching, defensive threat, defensive attack, and burying of potentially threatening objects (Blanchard and Blanchard, 2008; Blanchard et al., 1986; Blanchard et al., 1998). The category of response elicited by a given cue depends on the context in which it is presented (Bolles and Collier, 1976; Bouton and Bolles, 1980; Pinel and Triet, 1978; Yilmaz and Meister, 2013), the nature of the conditioned stimulus (Karpicke et al., 1977; Pinel and Triet, 1978), and the ongoing behavioral state of the animal (Fentress, 1968a, b).… The flight from danger, in contrast to freezing, is a stereotyped behavioral motif whose onset consists of a rapid transition in behavioral state (Blanchard et al., 1998; De Franceschi et al., 2016; Domenici and Blake, 1991; Walther, 1969; Yilmaz and Meister, 2013).” Thus, besides freezing, animals may increase locomotion when avoiding aversive stimuli such as foot shocks, when given the opportunity to escape them; in our experiment, the animals attempt to avoid the stimulus (Figures1A and B), which is likely the reason for increased locomotion. Therefore, we believe that these different aversive stimuli elicit comparable responses.

With regards to the reviewer’s final point, we are aware of this literature and had in fact discussed it in our manuscript, to illustrate that previous studies report dopamine-system responses in both directions: “Contradictory findings are also reported in the neighboring nucleus accumbens core (NAC), where studies found both increased (Budygin et al., 2012; Mikhailova et al., 2019) and decreased dopamine activity (Badrinarayan et al., 2012; Oleson et al., 2012; DeJong et al., 2019; Stelly et al., 2019).”. It is noteworthy that the experiments by Budygin et al. (2012) and by Mikhailova et al. (2019) were conducted in anesthetized animals, whereas the other experiments cited above were conducted in awake animals, as were ours.

EA Antoniadis, RJ McDonald. Amygdala, hippocampus, and unconditioned fear. Exp Brain Res. 2001 May;138(2):200-9. doi: 10.1007/s002210000645.

CE Schoonover, AJP Fink, R Axel (2017). A naturalistic assay for measuring behavioral responses to aversive stimuli at millisecond timescale. https://doi.org/10.1101/161885.

2. In Figure2A, the dopamine response to the white noise seems to be decreased. However, this is due to the baseline used to compute the white noise dopamine responses, which seems to shift lower due to the dopamine response to the antecedent cue. That is why the initial white noise dopamine response seems to be positive in Trial1 where the baseline is still above 0 but looks negative when the cue response becomes negative starting from Trial3. If the dopamine response to the white noise outcome were computed with a baseline of its own (1-2 sec before the WN outcome) that would result in a positive peak even in trials 11-30.In Figure 2A, the dopamine response to the white noise seems to be decreased.

Correct, this is what we report: dopamine decreases during WN presentation in Figure 2A, compared to baseline. Please see the traces and their respective bar graphs plotted over them, averaging the dopamine-response across 5s for baseline and cue, and 6s for WN responses.

However, this is due to the baseline used to compute the white noise dopamine responses, which seems to shift lower due to the dopamine response to the antecedent cue.

We believe this is a misunderstanding: the baseline used for the WN dopamine trace is the same as the baseline used for the cue trace (i.e., the 5s before cue onset (see Methods)). The average values of the dopamine traces for baseline, cue, and WN periods are depicted in the bar graphs inset on the plots (respectively white, light-grey, and dark-grey backgrounds).

That is why the initial white noise dopamine response seems to be positive in Trial1 where the baseline is still above 0 but looks negative when the cue response becomes negative starting from Trial3.

We understand that the reviewer observes by eye an increase in the initial dopamine response to WN in Trial1, however, the statistical analysis does not support this observation (there is no significant increase of dopamine during the first second of exposure to WN (p = 0.6606)) and, concomitantly, we did not report this as a result; the same is true for Trial3. Note: we would like to strongly emphasize that the error bars on the dopamine traces are substantial, which should be taken into consideration when translating graphical observations to statistical verification.

If the dopamine response to the white noise outcome were computed with a baseline of its own (1-2 sec before the WN outcome) that would result in a positive peak even in trials 11-30.

There was no significant positive peak in trials 11-30, no matter where the baseline for this trace is taken.

Somewhat unrelatedly: In case the reviewer assumes that the dopamine concentration stays decreased for the period of the variable inter-trial interval of 60s (range: 30-90 s): the five seconds following WN offset (displayed in Figure 2A) indicate that dopamine concentration trails back to baseline. This becomes more obvious in Figures 3D and E that depicts longer periods of time after WN offset.

3. The authors claim that the decrease in dopamine response to the white noise during the first trial of aversive conditioning (Figure 2A) is an unconditioned response. However, there is an immediate positive peak after the white noise presentation on that trial, which lasts about 1 sec. How does the behavior map on to this timeline? Do rats move for the first second but then freeze for the remainder of the white noise presentation? At the very least a strong justification should be made for what is being normalized and if and how you can separate specific task components.

As explained in our response to the reviewer’s point 2 (see above), this “immediate positive peak” is not statistically significant, and concomitantly, we did not report this as a result. Note: the error bars on the trial-one dopamine traces are substantial. Even in trial one, there is no significant increase of dopamine during the first second of exposure to WN (p = 0.6606) or its predictive cue (p = 0.0757). The initial bump in dopamine in Trial1 might be due to the novelty of the cue and the relative novelty of the first WN presentation of the session.

Hypothetically speaking, even if there were a statistically significant initial small peak in dopamine response (which there is not), we believe it would not seriously challenge our claim that the dopamine decrease to WN is unconditioned, since this hypothetical one-second dopamine bump is only present in Trial1. We do not think that it is reasonable to assume that a mild stressor like WN would lead to “conditioning” within one second.

Our data in Figure 1C (left) do not indicate any freezing. Furthermore, locomotion speed never decreases during the first trial of aversive Pavlovian conditioning (see Author response image 2 (n = 29)), instead we see an increase in speed during the WN epoch. Thus, WN induces an unconditioned response from the first exposure onwards. At no point in time do we see a significant drop in locomotion speed indicative of freezing behavior.

5. There are numerous studies where cues are paired with white noise as an aversive stimulus. It is important to determine if the predictive cue elicited a conditioned response. Without that how do you know the animals made the association? This is important to make conclusions about what the neural signal in response to these cues actually mean.

Although we do not agree with the reviewer that there are numerous studies where cues are paired with WN as an aversive stimulus, we completely agree that it is important to determine if the predictive cue elicited a conditioned response. To this end, we had already provided data that strongly supports this claim in Figure 2C (left) of our original version of the manuscript. In the revised version of the manuscript, we now add even more animals to the analysis in this figure panel. The result and its interpretation remain unchanged: The predictive cue elicits a conditioned response, as indicated by increased locomotion during cue presentation compared to baseline locomotion on day 6 of conditioning (Z = -3.053, p = 0.002), before the WN is presented (Figure 2C, left). Thus, we demonstrate that the animals made the association between cue and WN.

6. In response to the data with the white noise and different timing. Is this predicting the timing? Or the value of the outcome? These are not dissociable in this experiment and when you discuss timing this would be important to dissociate. This is a critical thing to parse as duration is in the title.

It is not 100% clear to us what the reviewer is referring to when stating: “Is this predicting the timing? Or the value of the outcome?”. We interpret the reviewer’s comment as asking whether our results indicate that dopamine tracks timing/duration or the value of the outcome. Which is an important point, a point that we addressed in our experiments. We concluded in the original manuscript that dopamine is “tracking the timing/duration” (i.e., presence of the WN), because we show that value was not encoded (Figures2E and G and Figure 3B), but that instead WN induces a categorical, slow and steady decrease in dopamine lasting the duration of the presence of the aversive stimulus (Figures3D-F), followed by a slow recovery period. The slope of this decrease is slower than it could be (compare to fast decreasing slope on Trial1 of Figure 2A), as is its recovery (compare to increase in dopamine in Figure 2B, top right). This, to us, conclusively demonstrates that the signal is tracking aversive duration, and that it is not tracking aversive value, as it is not conceivable to us how value could be represented in such a nearly binary fashion

Overall this is an interesting manuscript however in order for it to be suitable for publication the authors should rephrase their terminology to accurately state what the stimuli are and how they relate to behavior as well as make sure to show that white noise does function as an aversive stimulus.

We agree with the reviewer on terminology and have made appropriate changes throughout the manuscript.

…make sure to show that white noise does function as an aversive stimulus.

We demonstrated in two different experiments that WN is an aversive stimulus (Figures1A and B). Also, others have demonstrated the aversiveness of WN previously (e.g., Campbell 1955; Harrison and Tacy, 1955; Campbell and Bloom, 1965; Hughes and Bardo, 1981).

Campbell, BA (1955). The fractional reduction in noxious stimulation required to produce "just noticeable" learning. Journal of Comparative and Physiological Psychology, 48(3), 141–148.

Harrison JM, Tracy WH. Use of auditory stimuli to maintain lever-pressing behavior. Science. 1955 Mar 11;121(3141):373-4. 3.

Campbell, BA and Bloom, JM. Relative aversiveness of noise and shock. J. Comp. Physiol. Psychol. 60, 440–442 (1965).

Hughes, RA and Bardo, MT. Shuttlebox avoidance by rats using white noise intensities from 90-120 db SPL as the UCS. J. Aud. Res. 21, 109–118 (1981).

Reviewer #3 (Recommendations for the authors):In this study a talented group of neurochemists performed real-time measurements of dopamine concentration in behaving rats to investigate whether transient accumbal release events encode the value of aversive stimuli. Directly measuring dopamine release events rather than phasic bursts of putative dopamine neural activity is particularly important to determine how transient dopamine signals encode aversive events because recent evidence shows that terminal-terminal modulation influences behaviorally relevant patterns of release that do not necessarily coincide with changes in neural activity. The authors also incorporated an impressive systematic behavioral design and a unique aversive stimulus (i.e., white noise) to address this unresolved controversy. First, they determined that high decibel white noise produced a conditioned place aversion and punished food seeking. Then, by presenting comparable levels of white noise within a Pavlovian context, they found that dopamine release events were suppressed during the presentation an aversive stimulus and its conditioned predictor. They further report that the magnitude by which dopamine release events were suppressed did not correlate with the amplitude of white noise; thereby leading them to conclude that transient dopamine signals in the core region of the nucleus accumbens respond to aversive stimuli, but do not necessarily encode the value of punishment. However, there remain several unresolved issues and points of contention regarding the interpretation of the authors' results. Of note, they did not measure dopamine release during punished behavior, but rather in the presence of an aversive stimulus that increased the behavior being assessed. In addition, it is not clear whether the rats were able to discriminate between the tightly dispersed decibels of white noise presented during the Pavlovian task in which dopamine concentration was measured. While the current results are intriguing and a technical advance over preceding electrochemical studies, the overall picture of how transient dopamine signals throughout the mesocorticolimbic pathway encode aversive stimuli still requires further clarification that the current group of authors are capable of providing.The submission includes an excellent set of well-considered experiments; I am both impressed and intrigued. However, I do have some constructive criticism, suggestions, and alternative interpretations to consider.A timeline or illustration of the different subgroups and conditions under which FSCV recordings occurred would increase the readability of the manuscript.

We thank the reviewer for their suggestion, and have added Author response table 1 to the supplementary material of the manuscript:

**Author response table 1. sa2table1:** 

Experiment	N	FSCV readout	Behavioral readout
Real time place aversion of randomly assigned quadrant (Figure 1A)	10		X
Real time place aversion of preferred quadrant (Figure 1B)	15		X
Approach-avoidance foraging task (Figure 1C)	12		X
Operant box WN exposures (Figure 1D)	14		X
WN and reward choice task (Figure 1F)	6		X
Aversive Pavlovian conditioning day 1First 30 trials (Figure 2A, B, C and E)Mix trials (Figure 3D, E and F)	16 11	X X	X
Appetitive Pavlovian conditioning (Figure 2B and C)	10	X	X
Aversive Pavlovian conditioning days 2–6 (Figures 1E and 2B and C)	4 13	X	X X
Dose response (Figures 1D and 2F)	6 13	X	X X
Deterministic experiment (Figure 2E and G)	9	X	
Probabilistic experiment (Figures 2E, 3A and B and C)	9	X	
Concurrent reward & WN and extinction (Figures 2E and 3G and H)	6	X	

If the authors do not believe they are measuring dopamine (DA) value signals associated with aversive stimuli, have they considered whether they are measuring a correlate of the acoustic startle response? Acoustic startle is commonly associated with an increase in ambulation (as reported in the current manuscript). This alternative interpretation would provide an important missing piece of data from previously hypothesized neural circuitry underlying acoustic startle (see figure 6 of Koch and Schnitzler, 1997). Furthermore, the transient accumbal DA signal has previously been associated with pre-attentive sensory perception of salience. Might your results align more with Redgrave's work demonstrating that there are indeed distinct DA sensory responses that are dissociable from accumbal value signals; possibly also involved in the acoustic startle response?Koch M, Schnitzler HU. The acoustic startle response in rats-circuits mediating evocation, inhibition and potentiation. Behavioural brain research. 1997 Dec 1;89(1-2):35-49.

The reviewer suggests a reasonable hypothesis. The acoustic startle response (ASR) is one of the most commonly used behavioral responses to study habituation. In order to startle, ASR studies use sound intensities that are comparable (or higher) to the WN intensities we used. However, these studies commonly use a shorter sound duration of one second (we used six seconds) and a significantly lower number of trials per session (e.g., 10x), which are spaced apart with a shorter ITI than we used.

ASR habituation occurs both in the short-term and the long-term. More specifically, ASR short-term habituation occurs across trials within a session and long-term habituation across behavioral sessions. Thus, to study *short*-term habituation (a), amplitudes of the startle response across trials are compared, where a habituation effect occurs across several trials. To study *long*-term habituation (b), amplitudes of the startle response of the very first trial are compared across sessions, where a habituation effect occurs within five days.

We performed equivalent analyses on our dataset, comparing the average speed of our animals during WN exposures (a) across trials on day one and (b) in the first trials of sessions one and six:

a) Across trials on day one, we do indeed observe a decreased increase in locomotion speed during the WN epoch, which suggests that there might be short-term habituation (see Author response image 3). However, since this decrease in locomotion is not reflected in the dopamine responses (no differences in dopamine during the WN across these trials; see Figure 2A), we conclude that the dopamine responses that we observe are not encoding ASRs (under the assumption the reported behavioral response to WN is an ASR).

**Author response image 3. sa2fig3:** Locomotion speed across the first 30 trials on day 1 of conditioning of the rats (n=16) for which we show the dopamine traces in the NAC in Figure 2A.

b) When we compare locomotion speed during the WN epoch of the first trials on days one and six, we do not find a significant difference (t(16)=0.9111, p=0.3757) and, therefore, conclude that long-term habituation does not take place in our paradigm (see Author response image 4).

**Author response image 4. sa2fig4:** Comparing locomotion speed during the first trials of days 1 and 6 of aversive conditioning (total n = 17). No significant difference was found between average locomotion speed during the WN epoch in trials 1 between days 1 and 6 (t(16)=0.9111, p=0.3757), thus, no long-term habituation of the behavioral response occurred..

Together, these additional data do not support the interpretation that our reported dopamine signals encode an ASR to WN, because (1) the hypothetical long-term ASR does not habituate. We think it is more likely that our 6-s WN presentation (with longer ITIs) provokes a behavioral response that is inconsistent with startle. And (2) the hypothetical short-term ASR habituates, but this decrease in locomotion is not reflected in the dopamine signals. Furthermore, dopamine responses dopamine signals are not correlated with the behavioral response whatsoever.

Our results regarding negative dopamine signals in response to an aversive stimulus may, however, align with the idea of transient NAC dopamine signal associated with pre-attentive sensory perception of salience and, thus, with Redgrave's work demonstrating distinct dopamine sensory responses that are dissociable from NAC value signals.

The authors should discuss the current results in the context their previous work (specifically DA correlates with 22Kh USVs) with the Wohr lab, which was surprisingly not referenced in the current manuscript.Willuhn I, Tose A, Wanat MJ, Hart AS, Hollon NG, Phillips PE, Schwarting RK, Wöhr M. Phasic dopamine release in the nucleus accumbens in response to pro-social 50 kHz ultrasonic vocalizations in rats. Journal of Neuroscience. 2014 Aug 6;34(32):10616-23.

We did not reference our previous work on NAC dopamine responses to USVs in our manuscript because we did not think it relevant enough. Rats emit 22-kHz ultrasonic vocalizations (USVs) in aversive situations such as predator exposure, fear conditioning, or social defeat. Such calls are considered “alarm calls” that serve a communicative function, as their presentation induces call-specific freezing behavior in the receiver. WN on the other hand is frequency-unspecific noise (thereby essentially blocking the entire auditory modality, a sense that rodents rely on heavily), serves no communicative function, and, as a mildly aversive stimulus, does not induce freezing. Thus, WN does not induce the same behavioral response as 22-kHz (and probably does not elicit 22-kHz USVs). Taken together, it is not too surprising that qualitative differences are found in dopamine responses to the presentation of WN and 22-kHz USVs, as well as between 50-kHz (‘positive’ USVs) and 22-kHz USVs. On request of the reviewer, we have added a sentence on this matter to the Discussion section.

It is not clear whether rats could actually discriminate between the different tightly dispersed white noise volumes in the Pavlovian task during FSCV measurements-a centrally important experiment to support the authors' conclusions about dopamine and value. Based on my current interpretation of the methods, rats were able to discriminate between the white noise volumes in the operant choice task, but the conditions were substantially different from those in which the FSCV recordings occurred. Aside from being instrumental rather than Pavlovian, five consecutive training sessions occurred for each of three white noise volumes (70, 90, 96db) before discrimination testing was tested; and approximately 100 trials occurred in each session, with a single white noise volume being tested per session. Then, during volume-response FSCV recordings, each of four white noise volumes (70, 80, 90, 96db) were randomly played 12 times each in a single session. In a separate group of recordings, the authors performed FSCV recordings in the presence of 80 vs. 96db white noise volumes. Compared to the volume-response FSCV recordings, more training and trials occurred but, the Pavlovian trials were still randomly presented within a session. Thus, I caution against assuming that the animals could discriminate between the less dispersed and randomly presented white noise volumes presented in the single session Pavlovian experiments (particularly those depicted in 2E) based on the results from a methodologically distinct instrumental choice experiment. The authors should further address (either experimentally or logically) why the reader should accept that the rats could discriminate between the tightly spaced decibel volumes (particularly in the volume-response experiment).

The reviewer asks an important question: whether rats could discriminate between the different WN intensities. Naturally, we asked ourselves this question, too. First, we should correct a misunderstanding: these intensities are not “tightly dispersed” or “tightly spaced”, as decibels are on a logarithmic scale, and a 10dB difference equals a 10fold increase in intensity (perceived as a 2-fold increase in sound volume). This is easiest explained with real-life examples of the extremes of our range: 70dB is comparable to the sound of a running shower or a flushing toilet, whereas 96dB is the sound of a jackhammer or a power drill. Rats are well-hearing animals that rely on this sensory modality for many things. Based on appropriate literature, we chose our loudness range carefully, making sure that it was large enough for meaningful discrimination, and that the loudest condition is not harmful to the animals.

As pointed out by the reviewer, there was indeed more training leading up to the choice test, however, for the Pavlovian exposure, we also administered many trials: for the deterministic experiment (where 80 and 96 dB WN followed a cue with a 100% probability), FSCV measurements were carried out on the fourth day of conditioning (with 80 WN exposures per day) and for the probabilistic experiment (where 80 and 96 dB WN followed a cue with a probability of 75% or 25%), FSCV measurements were carried out after an additional four days of conditioning (with 80 WN exposures per day).

As the reviewer mentions, we show that the rats were able to discriminate between the WN intensities in the operant choice task. We disagree with the reviewer that the conditions in the operant choice task (where rats readily discriminated between white noise intensities), were substantially different from the Pavlovian presentation (where the reviewer questions the rats’ ability to discriminate between the WN intensities): Both were carried out in the exact same operant box, noise was played from the same speakers therein, and animals had the same amount of habituation. Thus, we believe that the WN intensities were perceived identically across these experiments, and it is therefore reasonable to assume that since the rats could discriminate between them in one, they could discriminate between them in the other.

Finally, we present new behavioral data in Figure 2F, which demonstrate that rats respond differently to different WN intensities, and thus discriminate between them.

Please provide additional clarification on figure 2B and your interpretations of it. First, the dopamine response to the CS predicting the aversive stimulus does seem to increase with experience (albeit quickly), which I contend contradicts your statement starting on line 293: 'Although the WN-predictive, conditioned cue acquired the ability to reliably suppress NAC dopamine release, no substantial transfer of this effect from US to CS occurred a prerequisite for a prediction error signal.' Aversive stimuli are known to rapidly induce conditioned responses. For example, a conditioned fear response is often established in a single fear conditioning session with just a few pairings of the CS and aversive stimuli. Thus, I contend (as the authors generally indicate in the results) that the US to CS transfer does indeed occur during aversive conditioning, it just occurred rapidly (on day 1; as would occur in standard fear conditioning). Similar to the presentation of extinction data in 2A, what do the CS and US data look like trial-by-trial on day 1 of aversive conditioning?

We agree with the reviewer, the data in Figure 2B could be described more clearly. We added information and clarification in response to the following remark by the reviewer (see next point on degrees of freedom).

…the dopamine response to the CS predicting the aversive stimulus does seem to increase with experience (albeit quickly), which I contend contradicts your statement starting on line 293: 'Although the WN-predictive, conditioned cue acquired the ability to reliably suppress NAC dopamine release, no substantial transfer of this effect from US to CS occurred a prerequisite for a prediction error signal.’

We are in agreement that some transfer does indeed occur during aversive conditioning. We do not state that “no transfer” occurs, we state that “no substantial transfer” occurs. This is an important distinction, as it implies that learning and associated dopamine-signal transfer occurs during aversive conditioning, however, with the distinction that only a relatively small percentage of the dopamine signal transfers from US to CS. (1) This percentage is small in comparison to appetitive conditioning; and (2) equally importantly, this percentage is not anywhere close to the amount of transfer that is a prerequisite for a prediction-error signal is (which requires full transfer with full predictability).

Similar to the presentation of extinction data in 2A, what do the CS and US data look like trial-by-trial on day 1 of aversive conditioning.

We are confused by this request. Figure 2A depicts “trial-by-trial data on day 1 of aversive conditioning” (some trials have been pooled because data does not change noticeably). Furthermore, Figure 2D depicts extinction data, not Figure 2A.

Please provide an explanation regarding the degrees of freedom associated with the statistics used for your comparisons in figure 2B (lines 242-244). It is unclear to me how the data points from the top figures transfer to the bottom figures and the degrees of freedom are adding to my confusion.

In order to make our point regarding the differences in the shift of the dopamine response from US to CS between appetitive and aversive Pavlovian conditioning more clear and easier to understand, we have now quantified this shift in a different manner than in the original manuscript (using the same data): In the new manuscript, we quantify this shift by calculating the ratio between US and CS dopamine concentration as a deviation from baseline (in the respective up or down direction). For aversive conditioning, this ratio was determined by dividing the area above the curve of the WN epoch by area above the curve of the cue epoch (WN AUC / cue AUC). For appetitive conditioning, this ratio was determined by dividing the area under the curve of the pellet epoch by the area under the curve of the cue epoch (pellet AUC / cue AUC).

We subdivided the results of day 1 into “day 1 (early)” (trials 2-4; trial 1 was excluded to minimize the saliency response) and “day 1 (late)” (trials 5-30), and also calculated the ratio for day 6. We used mixed effects analyses (because of one missing value for both aversive and appetitive Pavlovian conditioning) to separately test for the differences between the ratios during aversive and appetitive Pavlovian conditioning. During aversive conditioning we found a significant main effect of the amount of conditioning (F(0.9586, 2.396)=117.3, p = 0.0043), and post-hoc testing using Tukey’s multiple comparisons test reveals significant differences between the ratios of day 1 early trials and day 1 later trials (p = 0.0138), as well as between day 1 early trials and day 6 (p = 0.0318). However, no difference was observed between day 1 later trials and day 6 (p = 0.9852). During appetitive conditioning we also found a main effect of the amount of conditioning on the ratio between the US and the CS (F(1.034,8.788)=13.88, p=0.0047), and, in contrast to aversive conditioning, the ratio on day 6 is significantly different from both day 1 early trials (p=0.0102) and day 1 later trials (p<0.0001). In addition, we found a significant difference between day 1 early trials and day 1 later trials (p=0.0441). We have added this information to the manuscript.

Also, what affect did smoothing the data with a 10-point median filter prior to analysis have on the results? While smoothing the data for visual presentation is common, I question whether performing statistical analysis on the data after smoothing it might affect the results?

Smoothing data with 5- to 10-point filters is common practice for this type of data (including for statistics). It provides a better signal-to-noise ratio. To illustrate that this smoothing had no distorting effect on the data, we include a figure that compares raw data to the same data after being processed with the 10-point median filter (see Author response image 5). We can conclude from this graph that the smoothing of the data does not change the overall shape of the dopamine responses, but just eliminates the noise that accompanies this type of measurement technique.

**Author response image 5. sa2fig5:** Statistical effects are not distorted by 10-point median filter.

Furthermore, the maximal amplitude of the CS associated DA response during appetitive conditioning appears to be comparable between day 1 and day 6 in the top right panel of 2B. Thus, it seems that the duration of the signal, rather than its amplitude, is responsible for the significant effect shown in the bar graph of the bottom panel of 2B. From this observation, is the duration of the signal not accounting for the majority of appetitive value coding? Did smoothing the data contribute to the longer duration?From this observation, is the duration of the signal not accounting for the majority of appetitive value coding?

Although there is a change in the duration of the appetitive CS signal between days 1 and 6, there are a number of qualitative differences of this duration change compared to aversive conditioning (if that is what the reviewer is hinting at): (1) We know from a number of studies (including our own) that this average or AUC dopamine signal is highly responsive to changes in reward value (including changes in both amplitude and duration of the signal – as opposed to the WN response). (2) Increased release of dopamine (amplitude) automatically prolongs signal duration, since the signal is (mostly) actively inactivated (e.g., reuptake) – unlike decreases associated with WN (release could theoretically bring dopamine concentration back to baseline very quickly). (3) As can be seen from Figure 2B bottom right: The appetitive signal shift (US to CS) continues to grow dramatically within the session on day 1, as well as between days 1 and 6 (which includes a large amplitude change – unlike aversive conditioning). (4) The slope of change for dopamine concentration is different between appetitive and aversive conditioning. (5) See further points made in the discussion paragraph that compares appetitive and aversive conditioning.

Together, these points indicate that for appetitive conditioning, both duration and amplitude of the dopamine signal are modulated depending on the stimulus, whereas for aversive conditioning only the duration is modulated (depending on the stimulus duration).

Did smoothing the data contribute to the longer duration?

No, smoothing did not contribute to the duration of our dopamine signals. We used a 10-point median filter on data that was acquired with a rate of 10 Hz. Since we smoothed each data point by taking the median value of the 10 surrounding points (thus in total a time window of 1 second), the signal can only be prolonged for a maximum of 0.5 second. In the figure that accompanies our answer to the previous question, you can see an example graph in which we compared data which was processed with the 10 point median filter to the raw data. We can conclude from this graph that the smoothing of the data does not change the overall shape of the dopamine responses but just eliminates noise that is accompanied by this type of measuring technique.

The authors found that a longer duration of white-noise exposure produced a longer suppression in dopamine release. One could argue that both the frequency and amplitude of a signal should influence neural coding. Thus, why would a prolonged reduction in frequency not be reflective of greater aversive value? Why do the authors exclusively consider the amplitude of their dopamine in the context of aversive value determinations?

In our manuscript, we basically argue that in order for a brain signal to be a value signal, it should reflect all changes in value of a stimulus. Such changes in value include both the duration and the intensity of the “appraised” stimulus. In the case of WN, WN intensity is not encoded in NAC dopamine signals as Figure 2E and Figure 2G (formerly 2F) show clearly (all WN intensity produce the same amplitude in dopamine dip), even though the animals discriminated between intensities behaviorally. This is also true for Figures3BandC, where different probabilities alter the value of the stimuli, but do not alter the dopamine signal. So, we believe that this is already sufficient to disqualify NAC dopamine as encoding aversive value, since a value signal should reflect all general changes in the value of a stimulus (and it is not encoding an obvious, salient one such as intensity).

Even though we consider the above point to be sufficient for our argument, we will speculate with the reviewer on his hypothesis about duration: As we understand it, the reviewer is arguing that (despite the amplitude of the dopamine signal) value may be encoded in the dopamine signal in a quantitative manner by the duration of the dopamine dip. Our interpretation is that the dopamine dip is indeed encoding aversiveness, but in a qualitative manner (i.e., this is what we mean by tracking negative valence) via a constant decrease in dopamine during the presence of WN, because this constant decrease in the presence of WN is not modulated by WN intensity. In case duration of the dopamine dip encodes aversive value this way, it should drop for a differently long period of time with differently aversive WN, but we don’t see such duration encoding of value in Figure 2E, Figure 2G (formerly 2F), and Figures3B and C. Additionally, when unexpected WN durations are presented (Figures3D and E), the dip and recovery slopes are the same compared to the expected WN duration. Thus, the signal is not reacting to changes in WN duration in any other way than just decreasing in the presence of WN and increasing in the absense of WN (with the exception of Figure 3F).

The use of white noise as an aversive stimulus is championed by the authors because it does not produce the behavioral confound of freezing observed in standard fear conditioning approaches using electrical footshock. However, white noise and its conditioned predictor increased ambulatory behavior in the current study. Thus, how is the logic of avoiding behavioral DA responses not flawed? Could DA responses correlate to the initiation of action (which could be dissociable from speed) not confound DA value coding assessments across different volumes of white noise?

The reviewer raises an interesting hypothetical point, if there was any evidence for increased ambulatory behavior corresponding to decreased dopamine transients, as we see in our data. However, to our knowledge, all relevant literature that links dopamine to action finds these variables to correlate positively (i.e., there are no reported observations of decreasing dopamine-neuron activity). Knowing this, we believe the most obvious interpretation of our statement that using WN overcomes the common confound of freezing in foot shock approaches, is that we may be certain in our data that the dopamine decrease we observe is not due to lack of ambulatory behavior, but instead due to the aversive stimulus itself.

I also have some related critiques to consider regarding the benefits of white noise espoused in the discussion, starting on line 435. I already pointed out that the logic of using white noise because it doesn't induce freezing and thereby avoids movement-DA related confounds is flawed as white noise increased ambulation-which again could confound DA value coding of aversive stimuli if indeed DA transients are directly related to movement. But the data from the current study and others (PMID: 24345819, figure 2) might suggest that accumbal transients are not actually correlated in a positive way to general increases in activity. Regardless, I also take issue with point 2 and 3 (line 438) because foot shock can also maintain avoidance across many trials and sessions and be titrated with varying valence and contingencies; what are you contrasting white noise too? Point 4 does not remove the potential confound of Redgrave's work, as he has repeatedly demonstrated that the accumbal transient DA response can be induced by 'pre-attentive' subcortical sensory input. Point 6 is also not exactly accurate as properly isolating the electrical components of a shock generator eliminates the noise artifacts it can produce; thus, white noise might be easier but I don't think it is fair to imply that properly set-up electrical foot shock interferes with or jeopardizes FSCV recordings. It is fair that electrical artifacts are detected during single-unit recordings but they are easy to detect and remove during sorting.

We have addressed the reviewer’s first point with regards to locomotion as a confounding factor in our response to the previous point.

Regarding the reviewer taking issue with points 2 and 3: As stated by us on line 433 (of the previous manuscript draft), we are contrasting WN to more commonly-used aversive stimuli. This includes electric footshock, as well as other stimuli such as tail pinch, fox urine, social defeat, restraint/immobilization stress etc. Although we agree that electric shock is easier to titrate than most commonly-used aversive stimuli, we disagree that electric footshock at intensities that are commonly used is well-tolerated across many hundreds of trials. This is also the opinion of the Animal-research Ethics Committee we are working with. And as pointed out by reviewer 1, electric shock often involves pain.

And as stated by others (doi: 10.1016/j.bbr.2010.06.020): “.. electric shock has its downsides: It often induces secondary effects such as long term sleep disruption [41], altered social behavior [27], reduction in locomotion, rearing, and grooming behaviors, as well as an increase in immobility and defecation [45].” – Thus, it is beneficial to explore other aversive stimuli, as we have done here.

Furthermore, commonly-used metal grids for electric foot-shock delivery frequently fail to deliver shock reliably and/or consistently due to several factors (addressing point 3).

Regarding the reviewer’s comment on point 4: First of all, this list of advantages of WN is not restricted to the use of WN in dopamine research. Furthermore, it is very possible that an animal does not detect a stimulus light because it is turned away from the light source or has closed its eyes or has buried its snout in its fur while grooming. In all of these situations, WN would still be detected.

Regarding the reviewer’s comment on point 6: We have toned down this statement about electrical interference by removing the term “jeopardize”. However, at the very least, the data collected during foot-shock application is unusable (and in FSCV recordings often more than that), which is a period of significant value in a study such as ours. We are contented to hear the reviewer has no problems dealing with electrical interference from foot shocks in FSCV and single-unit data, as this is a prohibitive drawback in most labs that specialize in these types of recordings. This opinion is also shared by researchers that are using techniques other than FSCV: “Finally, the nature of footshock stimulation precludes its full inclusion in some modern experimental techniques, such as electrophysiology.” (doi: 10.1016/j.bbr.2010.06.020).

None of these points are intended to discredit foot shock as an aversive stimulus, which is obviously a very good model that has been and still is used widely, but merely to point out the benefits of WN.

The study would be strengthened by a core vs. shell vs. PFC comparison but at the very least, the literature regarding the role of dopamine in these distinct regions should be addressed in the discussion. I acknowledge that performing measurements of DA release in the PFC is a fraught endeavor but, at this level of journal I would expect that you at least address the Tye lab's data on aversive stimuli and the PFC (generally reviewed in: Vander Weele et al. 2019). Is it possible that DA signals in the PFC but not the NAc core encode aversive value? You also point out that Badrinarayan et al., 2012 reported that the same CS that reduced dopamine in the core increased DA in the shell-a somewhat paradoxical finding that was neither explored in the current study nor addressed in the discussion. While it is an important replication to show that NAc DA in the core is reduced by conditioned predictors of aversive stimuli, and the analysis done in figure 3 is an impressive advance over previous studies, I am left questioning the advance provided by the current data set. The primary positive effect in the core is a replication. Building a story on the general role transient dopamine signals play in encoding aversive stimuli using negative FSCV effects that were determined using tightly spaced decibels of white noise that the rat may not have been able to discriminate is shaky.Vander Weele CM, Siciliano CA, Tye KM. Dopamine tunes prefrontal outputs to orchestrate aversive processing. Brain research. 2019 Jun 15;1713:16-31

Although we agree with the reviewer that comparing our findings between accumbal subregions (including the shell and different dorso-striatal regions) and other, cortical regions is very interesting, this was explicitly not our question; we motivate our choice of brain region (NAC) in the introduction. Based on previous findings, the NAC was the most likely brain region to express a dopamine APE. The mentioned work by Vander Weele et al. (2019) on the effects of dopamine in the PFC on signal-to-noise ratio is very impressive and interesting, but it does not address APEs at all, which is why we did not discuss it in our manuscript.

We appreciate the reviewer’s question: “Is it possible that DA signals in the PFC but not the NAc core encode aversive value?” Yes, we believe that is very much possible, and indeed a very good point that is highly interesting and deserves the attention of future studies (especially, because anatomical findings indicate that different midbrain dopamine neurons project to NAC and cortex). But again, discussing this would be outside of the scope of our manuscript, since we do not have any experimental data to base such a discussion on. We believe that the best we can do in our manuscript is to acknowledge that our data suggest that aversive value and APEs are encoded in other brain regions (as mentioned in our discussion).

We believe it is more than just a replication to show that NAc DA in the core is reduced by conditioned predictors of aversive stimuli, in a field of research where results on this topic have been all over the place. We provide recordings that were conducted in several different paradigms that consistently improve clarity on this topic.

We have addressed in great detail the reviewer’s misunderstanding about the intensities of WN in a previous question, but to reiterate: these intensities are in no way tightly spaced. And, we have addressed the reviewer’s concern that rats may not have been able to discriminate the different WN intensities.

I strongly suggest that the authors not use the term punishment in the context of their results and instead use a term such as aversive stimulus. While multiple definitions of punishment exist in the literature, the one that is almost universally taught in the psychological context of animal behavior today is that of Azrin and Holz (1966), which considers punishment as a reduction in behavior in response to a stimulus. According to this definition, the behavioral response is the key element in determining whether you are observing punishment or a reinforcement; with punishment describing a decrease in behavior and reinforcement describing an increase in behavior. It is additionally worth considering that an aversive stimulus (e.g., electrical shock) can function as either a punisher or a reinforcer (i.e., something that increases behavior) in the operant context (McKearny, 1966; Morse and McKearney 1977), during imprinting (Hess, 1959), or in human interaction (Mello 1978; Sack and Miller, 1975). In the current study white noise does seem to punish food-maintained responding in one context, but also increases ambulatory behavior in another context.

We agree with the reviewer, and have made appropriate changes throughout the manuscript.

Along this same line of thought, could it not be concluded that DA might scale with the value of punishment but not of aversive stimuli that do not actually reduce the occurrence of a behavior? Punishment-associated DA signals in the basal ganglia might be more correlated to stimuli that actually reduce behavior given this neural circuit's well-determined role in goal-directed learning. Thus, if you want to conclude that DA fails to encode punishment (rather than an aversive stimulus), I would want to actually see transient DA signals failing to correlate to stimuli that reduce behavior to different magnitudes.

The reviewer raises a very good point, however we have followed his and the other reviewer’s rightly-made suggestion and replaced the term “punishment” throughout the manuscript.

[Editors’ note: what follows is the authors’ response to the second round of review.]

During the review process, the editors and reviewers evaluated and discussed your revised manuscript and your responses to the initial concerns. Unfortunately, all agreed that the data provided are insufficient to convince that the white noise is aversive. The data in Figure 1A and B that are used to make this point do not have the necessary controls. Further, db do not scale linearly. Therefore, the level of aversiveness of the different db of white noise also needs to be verified behaviourally. The valence of the white noise is a the backbone to the story of the paper and the absence of strong evidence that speaks to this issue within the paper was judged to be problematic, precluding it from further consideration for publication. We’re sure this is not the decision you were hoping for, but appreciate the chance to reconsider your manuscript and hope that our evaluation will be useful for you as you move forward with this work.

To address the above-mentioned concerns, we have added multiple new experiments to the current submission. For example, to improve the real-time place-aversion paradigm, we conducted an experiment that introduces white noise (WN) to the chamber quadrant that was previously preferred by the animals. Furthermore, we invented an approach-avoidance paradigm, which demonstrates that the utilized WN intensities do, in fact, scale linearly in their aversiveness. Repeated WN exposures in this paradigm demonstrate that the rats’ response to the different WN intensities were reliable across sessions, validating the absence of sensitization or habituation of the behavioral response to WN across sessions and days. Similarly, we now also demonstrate that (just like the behavioral response) the dopamine response to WN does not change across days.

Additionally, we control for the type of stimulus by introducing a 70-dB tone to the animals, which proved to be less aversive than 70-dB WN. Moreover, we provide data that shows how intensity-dependent and reliable across days the locomotor response to WN is. Together, we strongly believe, the results from these additional experiments address all remaining concerns raised after the previous article submission, strengthening the conclusions we drew in our previous submission.

[Editors' note: further revisions were suggested prior to acceptance, as described below.]

The manuscript has been improved but there are some remaining issues that need to be addressed, as outlined below:Reviewer #2 (Recommendations for the authors):The authors have done a good job at responding to previous comments. I do think that the study is interesting and important for the field. I have a few additional comments that should be addressed.Several manuscripts have come out recently specifically looking at dopamine release and aversive associative learning. These are surprisingly not cited or mentioned at all in the current manuscript and are highly relevant to the current work (Kutlu et al., 2022, Nature Neuroscience; Kutlu et al., 2021,. Current Biology). Both of these studies record dopamine release in the NAc core during aversive conditioning and relate dopamine signals to aversive stimulus responses and omissions based on previous predictions.

We agree with the reviewer that these references deserve being mentioned in the context of our work. Thus, as requested, we now cite these publications in the manuscript.

It would be interesting and important for the authors to discuss how aversive stimuli that induce different unconditioned responses – freezing, vs increased motor activity – could relate to dopamine signatures that they induce. Would the authors expect that dopamine responses to aversive stimuli that induce freezing be opposite to those that drive increases in activity?

The reviewer brings up a relevant point. As indicated by the editor, studies by others, including Robert Rescorla and Gavan McNally, utilized even louder white noise (around 120db) than we have in this work. We refrained from using WN at such intensities due to an elevated risk of loss of hearing (Escabi *et al.*, 2019), when used at durations that we applied in our experiments (6+ seconds per WN presentation). Brief 100-ms presentations of 120-dB WN induced fear responses such as freezing. Future studies will have to investigate whether such an opposite behavioral response (compared to behavioral activation reported here), is accompanied by differential dopamine dynamics, perhaps an increase in dopamine release (potentially related to increases in activity). Our analysis indicates that the NAC dopamine response is somewhat independent of behavioral actions (see Figure 2H). Thus, we speculate that despite opposite behavioral patterns, 120-dB WN will also induce decreased dopamine. This speculation is supported by the work of others, who found decreased dopamine release during the administration of electric shocks that induce freezing (Badrinarayan et al., 2012; Oleson et al., 2012; De Jong et al., 2019; Stelly et al., 2019).

We have added this point to the Discussion section of our manuscript.